# Hierarchical Sliced Wasserstein distance

**Khai Nguyen**
Department of Statistics and Data Sciences
The University of Texas at Austin
Austin, TX 78712
khainb@utexas.edu

**Tongzheng Ren**
Department of Computer Science
The University of Texas at Austin
Austin, TX 78712
tongzheng@utexas.edu

**Huy Nguyen**
Department of Statistics and Data Sciences
The University of Texas at Austin
Austin, TX 78712
huynm@utexas.edu

**Litu Rout**
Department of Electrical and Computer Engineering
The University of Texas at Austin
Austin, TX 78712
litu.rout@utexas.edu

**Tan Nguyen**
Department of Mathematics
University of California, Los Angeles
Los Angeles, CA 90095
tanmnguyen89@ucla.edu

**Nhat Ho**
Department of Statistics and Data Sciences
The University of Texas at Austin
Austin, TX 78712
minhnhat@utexas.edu

## Abstract

Sliced Wasserstein (SW) distance has been widely used in different application scenarios since it can be scaled to a large number of supports without suffering from the curse of dimensionality. The value of sliced Wasserstein distance is the average of transportation cost between one-dimensional representations (projections) of original measures that are obtained by Radon Transform (RT). Despite its efficiency in the number of supports, estimating the sliced Wasserstein requires a relatively large number of projections in high-dimensional settings. Therefore, for applications where the number of supports is relatively small compared with the dimension, e.g., several deep learning applications where the mini-batch approaches are utilized, the complexities from matrix multiplication of Radon Transform become the main computational bottleneck. To address this issue, we propose to derive projections by linearly and randomly combining a smaller number of projections which are named *bottleneck projections*. We explain the usage of these projections by introducing *Hierarchical Radon Transform* (HRT) which is constructed by applying Radon Transform variants recursively. We then formulate the approach into a new metric between measures, named *Hierarchical Sliced Wasserstein* (HSW) distance. By proving the injectivity of HRT, we derive the metricity of HSW. Moreover, we investigate the theoretical properties of HSW including its connection to SW variants and its computational and sample complexities. Finally, we compare the computational cost and generative quality of HSW with the conventional SW on the task of deep generative modeling using various benchmark datasets including CIFAR10, CelebA, and Tiny ImageNet[1].

## 1 Introduction

Wasserstein distance (Villani, 2008; Peyré & Cuturi, 2020) has been widely used in applications, such as generative modeling on images (Arjovsky et al., 2017; Tolstikhin et al., 2018; litu Rout et al., 2022), domain adaptation to transfer knowledge from source to target domains (Courty et al., 2017; Bhushan Damodaran et al., 2018), clustering problems (Ho et al., 2017), and various other

---

[1]Code for experiments in the paper is published at the following link https://github.com/UT-Austin-Data-Science-Group/HSW.

applications (Le et al., 2021; Xu et al., 2021; Yang et al., 2020). Despite the increasing importance of Wasserstein distance in applications, prior works have alluded to the concerns surrounding the high computational complexity of that distance. When the probability measures have at most $n$ supports, the computational complexity of Wasserstein distance scales with the order of $\mathcal{O}(n^3 \log n)$ (Pele & Werman, 2009). Additionally, it suffers from the curse of dimensionality, i.e., its sample complexity (the bounding gap of the distance between a probability measure and the empirical measures from its random samples) is of the order of $\mathcal{O}(n^{-1/d})$ (Fournier & Guillin, 2015), where $n$ is the sample size and $d$ is the number of dimensions.

Over the years, numerous attempts have been made to improve the computational and sample complexities of the Wasserstein distance. One primal line of research focuses on using entropic regularization (Cuturi, 2013). This variant is known as entropic regularized optimal transport (or in short entropic regularized Wasserstein). By using the entropic version, one can approximate the Wasserstein distance with the computational complexities $\mathcal{O}(n^2)$ (Altschuler et al., 2017; Lin et al., 2019b;a; 2020) (up to some polynomial orders of approximation errors). Furthermore, the sample complexity of the entropic version had also been shown to be at the order of $\mathcal{O}(n^{-1/2})$ (Mena & Weed, 2019), which indicates that it does not suffer from the curse of dimensionality.

Another line of work builds upon the closed-form solution of optimal transport in one dimension. A notable distance metric along this direction is sliced Wasserstein (SW) distance (Bonneel et al., 2015). SW is defined between two probability measures that have supports belonging to a vector space, e.g, $\mathbb{R}^d$. As defined in (Bonneel et al., 2015), SW is written as the expectation of one-dimensional Wasserstein distance between two projected measures over the uniform distribution on the unit sphere. Due to the intractability of the expectation, Monte Carlo samples from the uniform distribution over the unit sphere are used to approximate SW distance. The number of samples is often called the number of projections that is denoted as $L$. On the computational side, the projecting directions matrix of size $d \times L$ is sampled and then multiplied by the two data matrices of size $n \times d$ resulting in two matrices of size $n \times L$ that represent $L$ one-dimensional projected probability measures. Thereafter, $L$ one-dimensional Wasserstein distances are computed between the two corresponding projected measures with the same projecting direction. Finally, the average of those distances yields an approximation of the value of the sliced Wasserstein distance.

Prior works (Kolouri et al., 2018a; Deshpande et al., 2018; 2019; Nguyen et al., 2021a;b) show that the number of projections $L$ should be large enough compared to the dimension $d$ for a good performance of the SW. Despite the large $L$, SW has lots of benefits in practice. It can be computed in $\mathcal{O}(n \log_2 n)$ time, with the statistical rate $\mathcal{O}(n^{-1/2})$ that does not suffer from the curse of dimensionality, while becoming more *memory efficient*[2] compared with the vanilla Wasserstein distance. For these reasons, it has been successfully applied in several applications, such as (deep) generative modeling (Wu et al., 2019; Kolouri et al., 2018a; Nguyen & Ho, 2022a), domain adaptation (Lee et al., 2019), and clustering (Kolouri et al., 2018b). Nevertheless, it also suffers from certain limitations in, e.g., deep learning applications where the mini-batch approaches (Fatras et al., 2020) are utilized. Here, the number of supports $n$ is often much smaller than the number of dimensions. Therefore, the computational complexity of solving $L$ one-dimensional Wasserstein distance, $\Theta(Ln \log_2 n)$ is small compared to the computational complexity of matrix multiplication $\Theta(Ldn)$. This indicates that almost all computation is for the projection step. The situation is ubiquitous since there are several deep learning applications involving processing high-dimensional data, including images (Genevay et al., 2018; Nguyen & Ho, 2022b), videos (Wu et al., 2019), and text (Schmitz et al., 2018).

Motivated by the low-rank decomposition of matrices, we propose a more efficient approach to project original measures to their one-dimensional projected measures. In particular, two original measures are first projected into $k$ one-dimensional projected measures via Radon transform where $k < L$. For convenience, we call these projected measures as *bottleneck projections*. Then, new $L$ one-dimensional projected measures are created as random linear combinations of the bottleneck projections. The linear mixing step can be seen as applying Radon transform on the joint distribution of $k$ one-dimensional projected measures. From the computational point of view, the projecting step consists of two consecutive matrix multiplications. The first multiplication is between the data matrix of size $n \times d$ and the bottleneck projecting directions matrix of size $d \times k$, and the second multiplication is between the bottleneck projecting matrix and the linear mixing matrix of size $k \times L$. Columns of both the bottleneck projecting directions matrix and the linear mixing matrix are sampled

---

[2]SW does not need to store the cost matrix between supports.

randomly from the uniform distribution over the corresponding unit-hypersphere. For the same value of $L$, we show that the bottleneck projection approach is faster than the conventional approach when the values of $L$ and $d$ are large.

**Contribution:** In summary, our main contributions are two-fold:

1. We formulate the usage of bottleneck projections into a novel integral transform, *Hierarchical Radon Transform* (HRT) which is the composition of Partial Radon Transform (Liang & Munson, 1997) (PRT) and Overparameterized Radon Transform (ORT). By using HRT, we derive a new sliced Wasserstein variant, named *Hierarchical sliced Wasserstein* distance (HSW). By showing that HRT is injective, we prove that HSW is a valid metric in the space of probability measures. Furthermore, we discuss the computational complexity and the projection complexity of the proposed HSW distance. Finally, we derive the connection between HSW and other sliced Wasserstein variants, and the sample complexity of HSW.

2. We design experiments focusing on comparing HSW to the conventional SW in generative modeling on standard image datasets, including CIFAR10, CelebA, and Tiny ImageNet. We show that for approximately the same amount of computation, HSW provides better generative performance than SW and helps generative models converge faster. In addition, we compare generated images qualitatively to reinforce the favorable quality of HSW.

**Organization.** The remainder of the paper is organized as follows. We first provide background about Radon Transform, Partial Radon Transform, and sliced Wasserstein distance in Section 2. In Section 3, we propose Overparameterized Radon Transform, Hierarchical Radon Transform, Hierarchical sliced Wasserstein distance, and analyze their relevant theoretical properties. Section 4 contains the application of HSW to generative models, qualitative results, and quantitative results on standard benchmarks. We then draw concluding remarks in Section 5. Finally, we defer the proofs of key results, supplementary materials, and discussion on related works to Appendices.

**Notation.** For $n \in \mathbb{N}$, we denote by $[n]$ the set $\{1, 2, \ldots, n\}$. For any $d \geq 2$, $\mathbb{S}^{d-1} := \{\theta \in \mathbb{R}^d \mid \|\theta\|_2 = 1\}$ denotes the $d$ dimensional unit sphere, and $\mathcal{U}(\mathbb{S}^{d-1})$ is the uniform measure over $\mathbb{S}^{d-1}$. We denote the set of absolutely integrable functions on $\mathbb{R}^d$ as $\mathbb{L}_1(\mathbb{R}^d) := \{f : \mathbb{R}^d \to \mathbb{R} \mid \int_{\mathbb{R}^d} |f(x)| dx < \infty\}$. For $p \geq 1$, we denote the set of all probability measures on $\mathbb{R}^d$ that have finite $p$-moments as $\mathcal{P}_p(\mathbb{R}^d)$. For $m \geq 1$, we denote $\mu^{\otimes m}$ as the product measure of $m$ random variables that follow $\mu$ while $A^{\otimes m}$ indicates the Cartesian product of $m$ sets $A$. The Dirac delta function is denoted by $\delta$. For a vector $X \in \mathbb{R}^{dm}$, $X := (x_1, \ldots, x_m)$, $P_X$ denotes the empirical measures $\frac{1}{m} \sum_{i=1}^m \delta(x - x_i)$. For any two sequences $a_n$ and $b_n$, the notation $a_n = \mathcal{O}(b_n)$ means that $a_n \leq C b_n$ for all $n \geq 1$, where $C$ is some universal constant.

## 2 BACKGROUND

We first review the definition of the Radon Transform. We then review the sliced Wasserstein distance, and discuss its limitation in high-dimensional setting with relatively small number of supports.

**Definition 1** (Radon Transform (Helgason, 2011)). *The Radon Transform* $\mathcal{R} : \mathbb{L}_1(\mathbb{R}^d) \to \mathbb{L}_1(\mathbb{R} \times \mathbb{S}^{d-1})$ *is defined as:* $(\mathcal{R}f)(t, \theta) = \int_{\mathbb{R}^d} f(x) \delta(t - \langle x, \theta \rangle) dx$. *Note that, the Radon Transform defines a linear bijection.*

**Sliced Wasserstein distance:** From the definition of the Radon Transform, we can define the sliced Wasserstein distance as follows.

**Definition 2** (Sliced Wasserstein Distance (Bonneel et al., 2015)). *For any $p \geq 1$ and dimension $d \geq 1$, the sliced Wasserstein-$p$ distance between two probability measures $\mu \in \mathcal{P}_p(\mathbb{R}^d)$ and $\nu \in \mathcal{P}_p(\mathbb{R}^d)$ is given by:*

$$SW_p(\mu, \nu) = \left(\mathbb{E}_{\theta \sim \mathcal{U}(\mathbb{S}^{d-1})} W_p^p\left((\mathcal{R}f_\mu)(\cdot, \theta), (\mathcal{R}f_\nu)(\cdot, \theta)\right)\right)^{\frac{1}{p}}, \tag{1}$$

*where $f_\mu(\cdot), f_\nu(\cdot)$ are the probability density functions of $\mu$, $\nu$ respectively, and $W_p(\mu, \nu) := \left(\inf_{\pi \in \Pi(\mu, \nu)} \int_{\mathbb{R}^d \times \mathbb{R}^d} \|x - y\|_p^p d\pi(x, y)\right)^{\frac{1}{p}}$ is the Wasserstein distance of order $p$ (Villani, 2008; Peyré & Cuturi, 2019). With slightly abuse of notation, we use $W_p(\mu, \nu)$ and $W_p(f_\mu, f_\nu)$ interchangeably.*

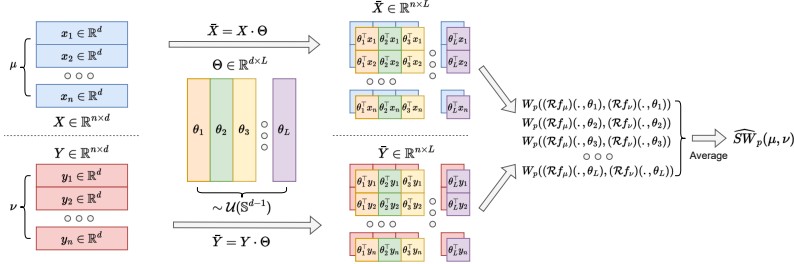

Figure 1: The Monte Carlo estimation conventional sliced Wasserstein distance with $L$ projections.

The main benefit of sliced Wasserstein is that the one-dimensional Wasserstein distance $W_p\left((\mathcal{R}f_\mu)(\cdot, \theta), (\mathcal{R}f_\nu)(\cdot, \theta)\right)$ has a closed-form $\left(\int_0^1 |F^{-1}_{((\mathcal{R}f_\mu)(\cdot, \theta)}(z) - F^{-1}_{(\mathcal{R}f_\nu)(\cdot, \theta)}(z)|^p dz\right)^{1/p}$, where $F^{-1}_{(\mathcal{R}f_\mu)(\cdot, \theta)}(z)$ is the inverse cumulative distribution function of the random variable that has the density $(\mathcal{R}f_\mu)(\cdot, \theta)$ (similarly for $F^{-1}_{(\mathcal{R}f_\nu)(\cdot, \theta)}(z)$). We denote the one-dimensional pushforward measures with the density $(\mathcal{R}f_\mu)(\cdot, \theta)$ as $\theta\sharp\mu$.

**Sliced Wasserstein distance between discrete measures:** In several applications, such as deep generative modeling (Deshpande et al., 2018; Nguyen & Ho, 2022b), domain adaptation (Lee et al., 2019), point cloud reconstruction (Nguyen et al., 2021c; 2023a), sliced Wasserstein distance had been used for discrete measures. Let two measures $\mu$ and $\nu$ that have the pdfs $f_\mu(x) = \frac{1}{n}\sum_{i=1}^n \alpha_i \delta(x - x_i)$, $f_\nu(y) = \frac{1}{m}\sum_{j=1}^m \beta_j \delta(y - y_j)$ ($\alpha_i, \beta_j > 0 \; \forall i, j$), and a given projecting direction $\theta \in \mathbb{S}^{d-1}$, the corresponding Radon Transform are $(\mathcal{R}f_\mu)(z, \theta) = \frac{1}{n}\sum_{i=1}^n \alpha_i \delta(z - \theta^\top x_i)$ and $(\mathcal{R}f_\nu)(z, \theta) = \frac{1}{m}\sum_{j=1}^m \beta_j \delta(z - \theta^\top y_j)$. Since the expectation in Definition 2 is intractable, Monte Carlo estimation is used with $L$ projecting directions $\theta_1, \ldots, \theta_L \sim \mathcal{U}(\mathbb{S}^{d-1})$:

$$\widehat{SW}_p(\mu, \nu) = \left(\frac{1}{L}\sum_{i=1}^L W_p^p\left((\mathcal{R}f_\mu)(\cdot, \theta_i), (\mathcal{R}f_\nu)(\cdot, \theta_i)\right)\right)^{\frac{1}{p}}. \tag{2}$$

We denote $X$ as the matrix of size $n \times d$ which has rows being supports of $\mu$, $[x_1, \ldots, x_n]^\top$, $Y$ as the matrix of size $m \times d$ which has rows being supports of $\nu$, $[y_1, \ldots, y_m]^\top$, and $\Theta$ as the matrix of size $d \times L$ which has columns being sampled projecting directions $[\theta_1, \ldots, \theta_L]$. The supports of the Radon Transform measures from $\mu$ and $\nu$ are the results of matrix multiplication $\bar{X} = X \cdot \Theta$ (with the shape $n \times L$) and $\bar{Y} = Y \cdot \Theta$ (with the shape $m \times L$). Columns of $\bar{X}$ and $\bar{Y}$ are supports of projected measures. Therefore, $L$ one-dimensional Wasserstein distances are computed by evaluating the quantile functions which are based on sorting columns. A visualization is given in Figure 1.

**Computational and projection complexities of sliced Wasserstein distance:** Without the loss of generality, we assume that $n \geq m$, the time complexity of sliced Wasserstein is $\mathcal{O}(Ldn + Ln\log_2 n)$ where $Ldn$ is because of matrix multiplication ($X \cdot \Theta$) and $Ln\log_2 n$ is because of the sorting algorithm. The projection complexity of SW is the memory complexity for storing the projecting directions $\Theta$ which is $\mathcal{O}(Ld)$. We would like to remark that, the value of number of projections $L$ should be comparable to the number of dimension $d$ for a good performance in applications (Deshpande et al., 2018; Nguyen et al., 2021a;b).

**Computational issuse of sliced Wasserstein when $d \gg n$:** In deep learning applications (Deshpande et al., 2018; Lee et al., 2019; Kolouri et al., 2018a) where mini-batch approaches are used, the number of dimension $d$ is normally much larger than the number of supports $n$, e.g., $d = 8192$ and $n = 128$. Therefore, $\log_2 n \ll d$ that leads to the fact that the main computation of sliced Wasserstein is for doing projecting measures $\mathcal{O}(Ldn)$. To the best of our knowledge, prior works have not adequately addressed this limitation of sliced Wasserstein.

## 3 HIERARCHICAL SLICED WASSERSTEIN DISTANCE

In this section, we propose an efficient way to improve the projecting step of sliced Wasserstein distance. In particular, we first project measures into a relatively small number of projections ($k$), named *bottleneck projections* ($k < L$). After that, $L$ projections are created by a random linear

combination of the bottleneck projections. To explain the usage of the bottleneck projections, we first introduce *Hierarchical Radon Transform* (HRT) in Section 3.1. We then define the *Hierarchical Sliced Wasserstein* (HSW) distance and investigate its theoretical properties in Section 3.2. We show that the usage of bottleneck projections appears in an efficient estimation of HSW.

## 3.1 HIERARCHICAL RADON TRANSFORM

To define the Hierarchical Randon Transform, we first need to review an extension of Radon Transform which is *Partial Radon Transform* (PRT). After that, we propose a novel extension of Radon Transform which is named *Overparameterized Radon Transform* (ORT).

**Definition 3** (Partial Radon Transform (Liang & Munson, 1997)). *The Partial Radon Transform (PRT)* $\mathcal{PR} : \mathbb{L}_1(\mathbb{R}^{d_1} \times \mathbb{R}^{d_2}) \to \mathbb{L}_1\left(\mathbb{R} \times \mathbb{S}^{d_1-1} \times \mathbb{R}^{d_2}\right)$ *is defined as:* $(\mathcal{PR}f)(t, \theta, y) = \int_{\mathbb{R}^{d_1}} f(x, y)\delta(t - \langle x, \theta \rangle)dx$. *Given a fixed $y$, the Partial Radon Transform is the Radon Transform of $f(\cdot, y)$.*

**Definition 4** (Overparameterized Radon Transform). *The Overparameterized Radon Transform (ORT) $\mathcal{OR} : \mathbb{L}_1(\mathbb{R}^d) \to \mathbb{L}_1\left(\mathbb{R}^{\otimes k} \times \mathbb{S}^{(d-1)\otimes k}\right)$ is defined as:*

$$(\mathcal{OR}f)(t_{1:k}, \theta_{1:k}) = \int_{\mathbb{R}^d} f(x) \prod_{i=1}^{k} \delta(t_i - \langle x, \theta_i \rangle)dx, \tag{3}$$

*where $t_{1:k} := (t_1, \ldots, t_k) \in \mathbb{R}^{\otimes k}$ and $\theta_{1:k} := (\theta_1, \ldots, \theta_k) \in (\mathbb{S}^{d-1})^{\otimes k}$.*

Definition 4 is called "overparameterized" since the dimension of the transformed function's arguments is higher than the original dimension. Our motivation for ORT comes from the success of overparametrization in deep neural networks (Allen-Zhu et al., 2019).

**Proposition 1.** *The Overparameterized Radon Transform (ORT) is injective, i.e., for any functions $f, g \in \mathbb{L}^1(\mathbb{R}^d)$, $(\mathcal{OR}f)(t_{1:k}, \theta_{1:k}) = (\mathcal{OR}g)(t_{1:k}, \theta_{1:k}) \ \forall t_{1:k}, \theta_{1:k}$ implies that $f = g$.*

Since ORT is an extension of RT, the injectivity of ORT is derived from the injectivity of RT. The proof of Proposition 1 is in Appendix C.1.

We now define the *Hierarchical Radon Transform* (HRT).

**Definition 5** (Hierarchical Radon Transform). *Hierarchical Radon Transform (HRT) $\mathcal{HR} : \mathbb{L}_1(\mathbb{R}^d) \to \mathbb{L}_1\left(\mathbb{R} \times \mathbb{S}^{(d-1)\otimes k} \times \mathbb{S}^{k-1}\right)$ is defined as:*

$$(\mathcal{HR}f)(v, \theta_{1:k}, \psi) = \int_{\mathbb{R}^d} f(x)\delta\left(v - \sum_{i=1}^{k} \langle x, \theta_i \rangle \psi_i\right)dx, \tag{4}$$

*where $v \in \mathbb{R}, \psi = (\psi_1, \ldots, \psi_k) \in \mathbb{S}^{k-1}$, and $\theta_{1:k} = (\theta_1, \ldots, \theta_k) \in (\mathbb{S}^{d-1})^{\otimes k}$.*

Definition 5 is called "hierarchical" since it is the composition of Partial Radon Transform and Overparameterized Radon Transform. We can verify that $(\mathcal{HR}f)(v, \theta_{1:k}, \psi) = (\mathcal{PR}(\mathcal{OR}f))(v, \theta_{1:k}, \psi)$ (we refer the reader to Appendix B for the derivation). To the best of our knowledge, ORT and HRT have not been proposed in the literature.

**Proposition 2.** *The Hierarchical Radon Transform (HRT) is injective, i.e., for any functions $f, g \in \mathbb{L}^1(\mathbb{R}^d)$, $(\mathcal{HR}f)(v, \theta_{1:k}, \psi) = (\mathcal{HR}g)(v, \theta_{1:k}, \psi) \ \forall v, \theta_{1:k}, \psi$ implies that $f = g$.*

Since HRT is the composition of PRT and ORT, the injectivity of HRT is derived from the injectivity of ORT and PRT. The proof of Proposition 2 is in Appendix C.2.

**Hierarchical Radon Transform of discrete measures:** Let $f(x) = \frac{1}{n}\sum_{j=1}^{n} \alpha_i \delta(x - x_j)$, we have $(\mathcal{HR}f)(v, \theta_{1:k}, \psi) = \frac{1}{n}\sum_{i=1}^{n} \alpha_i \delta\left(v - \sum_{j=1}^{k} \langle x_i, \theta_j \rangle \psi_j\right) = \frac{1}{n}\sum_{i=1}^{n} \alpha_i \delta\left(v - \psi^\top \Theta^\top x_i\right)$, where $\Theta$ is the matrix columns of which are $\theta_{1:k}$. Here, $\theta_{1:k}$ are bottleneck projection directions, $\psi$ is the mixing direction, $\psi^\top \Theta$ is the final projecting direction, $\theta_j^\top x_i, \ldots, \theta_j^\top x_n$ for any $j \in [k]$ are the bottleneck projections, and $\psi^\top \Theta^\top x_i, \ldots, \psi^\top \Theta^\top x_n$ is the final projection. We consider that the bottleneck projections are the spanning set of a subspace that has the rank at most $k$ and the final projection belongs to that subspace. Based on the linearity of Gaussian distributions, we provide the result of HRT on multivariate Gaussian and mixture of multivariate Gaussians in Appendix B.

**Applications of Hierarchical Radon Transform:** In this paper, we focus on showing the benefit of the HRT in the sliced Wasserstein settings. However, similar to the Radon Transform, the HRT can also be applied to multiple other applications such as sliced Gromov Wasserstein (Vayer et al., 2019), sliced mutual information Goldfeld & Greenewald (2021), sliced score matching (sliced Fisher divergence) Song et al. (2020), sliced Cramer distance (Kolouri et al., 2019a), and other tasks that need to project probability measures.

## 3.2 HIERARCHICAL SLICED WASSERSTEIN DISTANCE

By using Hierarchical Radon Transform, we define a novel variant of sliced Wasserstein distance which is named *Hierarchical Sliced Wasserstein* (HSW) distance.

**Definition 6.** *For any $p \geq 1$, $k \geq 1$, and dimension $d \geq 1$, the hierarchical sliced Wasserstein distance of order $p$ between two probability measures $\mu \in \mathcal{P}_p(\mathbb{R}^d)$ and $\nu \in \mathcal{P}_p(\mathbb{R}^d)$ is given by:*

$$HSW_{p,k}(\mu, \nu) = \left( \mathbb{E}_{\theta_{1:k}, \psi} W_p^p \left( (\mathcal{HR} f_\mu)(\cdot, \theta_{1:k}, \psi), (\mathcal{HR} f_\nu)(\cdot, \theta_{1:k}, \psi) \right) \right)^{\frac{1}{p}}, \tag{5}$$

*where $\theta_1, \ldots, \theta_k \sim \mathcal{U}(\mathbb{S}^{d-1})$ and $\psi \sim \mathcal{U}(\mathbb{S}^{k-1})$.*

**Properties of hierarchical sliced Wasserstein distance:** First, we have the following result for the metricity of HSW.

**Theorem 1.** *For any $p \geq 1$ and $k \geq 1$, the hierarchical sliced Wasserstein $HSW_{p,k}(\cdot, \cdot)$ is a metric on the space of probability measures on $\mathbb{R}^d$.*

Proof of Theorem 1 is given in Appendix C.3. Our next result establishes the connection between the HSW, max hierarchical sliced Wasserstein (Max-HSW) (see Definition 8 in Appendix B), max sliced Wasserstein (Max-SW) (see Definition 7 in Appendix B), and Wasserstein distance. We refer the reader to Appendix B for more theoretical properties of the Max-HSW. The role of Max-HSW is to connect HSW with Max-SW that further allows us to derive the sample complexity of HSW.

**Proposition 3.** *For any $p \geq 1$ and $k \geq 1$, we find that*

*(a) $\frac{1}{k} HSW_{p,k}(\mu, \nu) \leq \frac{1}{k} Max\text{-}HSW_{p,k}(\mu, \nu) \leq Max\text{-}SW_p(\mu, \nu) \leq W_p(\mu, \nu)$,*

*(b) $SW_p(\mu, \nu) \leq Max\text{-}SW_p(\mu, \nu) \leq Max\text{-}HSW_{p,k}(\mu, \nu)$,*

*where we define*

$$Max\text{-}HSW_{p,k}(\mu, \nu) := \max_{\theta_1, \ldots, \theta_k \in \mathbb{S}^{d-1}, \psi \in \mathbb{S}^{k-1}} W_p \left( (\mathcal{HR} f_\mu)(\cdot, \theta_{1:k}, \psi), (\mathcal{HR} f_\nu)(\cdot, \theta_{1:k}, \psi) \right),$$

$$Max\text{-}SW_p(\mu, \nu) := \max_{\theta \in \mathbb{S}^{d-1}} W_p \left( (\mathcal{R} f_\mu)(\cdot, \theta), (\mathcal{R} f_\nu)(\cdot, \theta) \right)$$

*as max hierarchical sliced $p$-Wasserstein, and max sliced $p$-Wasserstein, respectively.*

Proof of Proposition 3 is given in Appendix C.4. Given the bounds in Proposition 3, we demonstrate that the hierarchical sliced Wasserstein does not suffer from the curse of dimensionality for the inference purpose, namely, the sample complexity for the empirical distribution from i.i.d. samples to approximate their underlying distribution is at the order of $\mathcal{O}(n^{-1/2})$ where $n$ is the sample size.

**Proposition 4.** *Assume that $P$ is a probability measure supported on compact set of $\mathbb{R}^d$. Let $X_1, X_2, \ldots, X_n$ be i.i.d. samples from $P$ and we denote $P_n = \frac{1}{n} \sum_{i=1}^n \delta_{X_i}$ as the empirical measure of these data. Then, for any $p \geq 1$, there exists a universal constant $C > 0$ such that*

$$\mathbb{E}[HSW_{p,k}(P_n, P)] \leq Ck\sqrt{(d+1)\log n/n},$$

*where the outer expectation is taken with respect to the data $X_1, X_2, \ldots, X_n$.*

Proof of Proposition 4 is given in Appendix C.5.

**Monte Carlo estimation:** Similar to SW, the expectation in Definition 6 is intractable. Therefore, Monte Carlo samples $\psi_1, \ldots, \psi_L \sim \mathcal{U}(\mathbb{S}^{k-1})$ (by abuse of notations) and $\theta_{1,1}, \ldots, \theta_{1,H}, \ldots, \theta_{k,1}, \ldots, \theta_{k,H} \sim \mathcal{U}(\mathbb{S}^{d-1})$ are used to approximate the HSW, which leads to the following approximation:

$$\widehat{HSW}_{p,k}(\mu, \nu) = \left( \frac{1}{HL} \sum_{h=1}^H \sum_{l=1}^L W_p^p \left( (\mathcal{HR} f_\mu)(\cdot, \theta_{1:k,h}, \psi_l), (\mathcal{HR} f_\nu)(\cdot, \theta_{1:k,h}, \psi_l) \right) \right)^{\frac{1}{p}}. \tag{6}$$

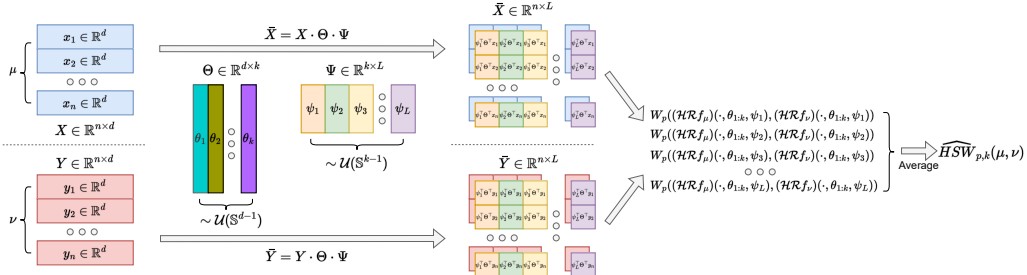

Figure 2: The Monte Carlo estimation of hierarchical sliced Wasserstein distance (HSW) with $H = 1$, $k$ bottleneck projections, and $L$ final projections.

**Computational and projection complexities on discrete measures:** It is clear that the time complexity of the Monte Carlo estimation of HSW with discrete probability measures of $n$ supports is $\mathcal{O}(Hkdn + HLkn + HLn\log_2 n)$. The projection complexity of HSW is $\mathcal{O}(Hdk + kL)$. For a fast computational complexity and a low projection complexity, we simply choose $H = 1$. In this case, the computational complexity is $\mathcal{O}(kdn + Lkn + Ln\log_2 n)$ and the projection complexity is $\mathcal{O}(dk + kL)$. Recall that $k < L$, which implies the estimation of HSW is faster and more efficient in memory than the estimation of SW. This fast estimator is unbiased, though its variance might be high.

**Benefit of HSW when $d \gg n$:** For the same computation complexity, HSW can have a higher number of final projections $L$ than SW. For example, when $d = 8192$ and $n = 128$ (the setting that we will use in experiments), SW with $L = 100$ has the computational complexity proportion to $104.94 \times 10^6$ and the projection complexity proportion to $0.82 \times 10^6$. In the same setting of $d$ and $n$, HSW with $H = 1, k = 50$ and $L = 500$ has the computational complexity proportion to $89.45 \times 10^6$ and the projection complexity proportion to $0.66 \times 10^6$.

**Implementation of HSW on discrete measures:** The slicing process of HSW contains $H$ matrices of size $d \times k$ in the first level and a matrix of size $k \times L$ in the second level. The projected measures are obtained by carrying out matrix multiplication between the support matrices with the projection matrices in the two layers in turn. When $H = 1$, we observe the composition of projection matrices in the two layers as a two-layer neural network with linear activation. However, the weights of the neural network have a constraint (spherical constraint) and are sampled instead of being optimized. In the paper, since we focus on the efficiency of the estimation, we consider only settings where $H = 1$. A visualization of the process when $H = 1$ is given in Figure 2.

**Beyond single hierarchy with standard Radon transform:** In HRT, PRT is applied on the arguments $(t_1, \ldots, t_k)$. By applying many PRTs on subsets of arguments e.g., $(t_1, t_2), \ldots, (t_{k-1}, t_k)$ and then applying PRT again on the arguments of the output function, we derive a more hierarchical version of HRT. Similarly, we could also apply ORT on the arguments multiple times to make the transform hierarchical. Despite the hierarchy, Hierarchical Sliced Wasserstein distance still uses linear projections. By changing from Radon Transform variants to Generalized Radon Transform (Beylkin, 1984) variants, we easily extend Hierarchical Radon Transform to Hierarchical Generalized Radon Transform (HGRT). Furthermore, we derive the Hierarchical Generalized Sliced Wasserstein (HGSW) distance. Since HGSW has more than one non-linear transform(layer), it has the provision of using a more complex non-linear transform than the conventional Generalized sliced Wasserstein (GSW) (Kolouri et al., 2019b). Compared to the neural network defining function of GRT (Kolouri et al., 2019b), HGSW preserves the metricity, i.e., HGSW satisfies the identity property due to the injectivity of HGRT with defining functions that satisfy constraints H1-H4 in (Kolouri et al., 2019b). Since the current set defining functions e.g., circular functions and homogeneous polynomials with an odd degree are not scalable in high-dimension, we defer the investigation of HGSW and finding a more scalable function for future work.

**Distributions of final projecting directions in HRT:** We recall that the final projecting directions of HSW are $\psi_1^\top \Theta, \ldots, \psi_L^\top \Theta$, where $\Theta = (\theta_{1:k})$, $\theta_{1:k} \sim \mathcal{U}(\mathbb{S}^{d-1})$, and $\psi_1 \ldots, \psi_L \sim \mathcal{U}(\mathbb{S}^{k-1})$. It is clear that the final projecting directions are distributed uniformly from the manifold $\mathcal{S} := \{x \in \mathbb{R}^{d-1} \mid x = \sum_{i=1}^{k} \psi_i^\top \theta_i, (\psi_1, \ldots, \psi_k) \in \mathbb{S}^{k-1}, \theta_i \in \mathbb{S}^{d-1}, \forall i = 1, \ldots, k\}$. Therefore, HSW can be considered as the sliced Wasserstein with the projecting directions on a special manifold. This is different from the conventional unit hyper-sphere. To our knowledge, the manifold $\mathcal{S}$ has not been explored sufficiently in previous works, which may be a potential direction of future research.

Table 1: Summary of FID scores, IS scores, computational complexity, memory complexity, computational time in millisecond (ms) of different estimations of SW and HSW on CIFAR10 (32x32), CelebA (64x64), and Tiny ImageNet (64x64).

| Method | Com ($\downarrow$) | Proj ($\downarrow$) | Time ($\downarrow$) | CIFAR10 | | CelebA | Tiny ImageNet | |
|---|---|---|---|---|---|---|---|---|
| | | | | FID ($\downarrow$) | IS ($\uparrow$) | FID ($\downarrow$) | FID ($\downarrow$) | IS ($\uparrow$) |
| GIS (Dai & Seljak, 2021) | - | - | - | 66.5 | - | 37.3 | - | - |
| SW (L=100) | 104.95 | 0.82 | 1 | 51.62±3.69 | 5.74±0.28 | **17.54±1.85** | 96.03±3.17 | 5.38±0.29 |
| HSW (k=70, L=2000) | **93.11** | **0.71** | 1 | **47.64±5.20** | **5.98±0.22** | 17.59±2.12 | **89.77±3.56** | **5.83±0.31** |
| SW (L=1000) | 1049.47 | 8.19 | 1.2 | 42.26±3.52 | 6.30±0.19 | 17.35±2.56 | 84.67±3.93 | 5.98±0.17 |
| HSW (k=400, L=6000) | **732.01** | **5.68** | 1.1 | **41.80±1.08** | **6.38±0.15** | **15.89±2.19** | **82.52±4.40** | **6.00±0.19** |
| SW (L=10000) | 10494.72 | 81.92 | 1.3 | 38.60±2.23 | 6.54±0.18 | 16.05±1.64 | 84.37±3.68 | 6.06±0.21 |
| HSW (k=3000, L=18000) | **10073.85** | **78.58** | 1.3 | **38.22±4.96** | **6.57±0.32** | **15.74±1.46** | **80.69±5.87** | **6.07±0.25** |

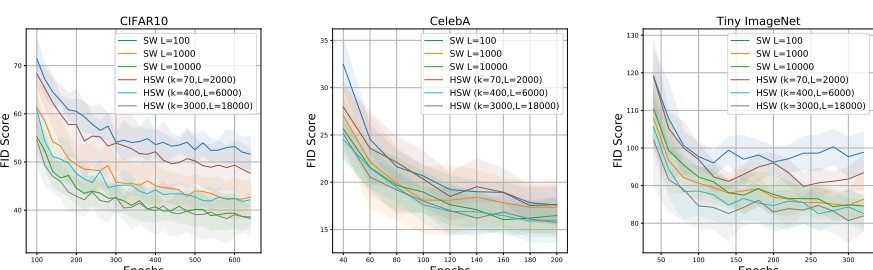

Figure 3: The FID scores over epochs of different training losses on datasets. We observe that HSW helps the generative models converge faster.

**On the choices of $k$ and $L$ in HSW:** We consider the HSW as an alternative option for applications that use the SW. For a given value of $L$ in the SW, we can choose a faster setting of HSW by selecting $k \leq \frac{Ld}{L+d}$ based on the analysis of their computational complexities. Similarly, the HSW with $L_2$ final projections can be faster than the SW with $L_1$ projections by choosing $k \leq \frac{L_1 d - (L_2 - L_1) \log_2 n}{d + L_2}$. Later, we use this rule for the choices of $k$ in the experiments while comparing HSW with SW.

## 4 EXPERIMENTS

In this section, we compare HSW with SW on benchmark datasets: CIFAR10 (with image size 32x32) (Krizhevsky et al., 2009), CelebA (with image size 64x64), and Tiny ImageNet (with image size 64x64) (Le & Yang, 2015). To this end, we consider deep generative modeling with the standard framework of the sliced Wasserstein generator (Deshpande et al., 2018; Nguyen et al., 2021a; Deshpande et al., 2019; Nguyen & Ho, 2022b; Nadjahi et al., 2021). We provide a detailed discussion of this framework including training losses and their interpretation in Appendix D.1. The SW variants are used in the feature space with dimension $d = 8192$ and the mini-batch size 128 for all datasets. The main evaluation metrics are FID score (Heusel et al., 2017) and Inception score (IS) (Salimans et al., 2016). We do not report the IS score on CelebA since it poorly captures the perceptual quality of face images (Heusel et al., 2017). The detailed settings about architectures, hyper-parameters, and evaluation of FID and IS are provided in Appendix F.

Our experiments aim to answer the following questions: **(1)** *For approximately the same computation, is the HSW comparable in terms of perceptual quality while achieving better convergence speed?* **(2)** *For the same number of final projections $L$ and a relatively low number of bottleneck projections $k < L$, how is the performance of the HSW compared to the SW?* **(3)** *For a fixed value of bottleneck projections $k$, does increasing the number of final projections $L$ improve the performance of the HSW?* **(4)** *For approximately the same computation, does reducing the value of $k$ and raising the value of $L$ lead to a better result?* After running each experiment 5 times, we report the mean and the standard deviation of evaluation metrics.

**The HSW is usually better than the SW with a lower computation:** We report the FID scores, IS scores, the computational complexity ($\times 10^6$), and the projection complexity ($\times 10^6$) for the SW with $L \in \{100, 1000, 10000\}$ and the HSW with $(k, L) = \{(70, 2000), (400, 6000), (3000, 18000)\}$ respectively in Table 1. The computational complexities and the projection complexities are computed based on the big O notation analysis of SW and HSW in previous sections. According to the table, the HSW usually yields comparable FID and IS scores while having a **lower computational complexity**

Table 2: FID scores, IS scores, computational complexity, and memory complexity for ablation studies of $k$ and $L$ of the SW on CIFAR10 (32x32), CelebA (64x64), and Tiny ImageNet (64x64).

| Method | Com ($\downarrow$) | Proj ($\downarrow$) | CIFAR10 | | CelebA | Tiny ImageNet | |
|---|---|---|---|---|---|---|---|
| | | | FID ($\downarrow$) | IS ($\uparrow$) | FID ($\downarrow$) | FID ($\downarrow$) | IS ($\uparrow$) |
| SW (L=1000) | 1049.47 | 8.19 | 42.26±3.52 | 6.30±0.19 | 17.35±2.56 | 84.67±3.93 | 5.98±0.17 |
| HSW (k=500, L=1000) | 589.18 | 4.59 | 43.58±4.01 | 6.25±0.31 | 17.50±2.25 | 89.02±2.32 | 5.92±0.19 |
| HSW (k=500, L=4000) | 783.87 | 6.09 | 41.85±4.42 | 6.36±0.23 | 16.80±1.23 | 86.57±3.80 | 5.94±0.45 |
| HSW (k=400, L=6000) | 732.01 | 5.68 | 41.80±1.08 | 6.38±0.15 | 15.89±2.19 | 82.52±4.40 | 6.00±0.19 |
| HSW (k=100, L=50000) | 789.65 | 5.81 | 44.70±3.19 | 6.09±0.15 | 17.41±2.12 | 89.01±2.73 | 5.90±0.21 |

and a **lower projection complexity** on benchmark datasets. We show random generated images on CIFAR10 in Figure 5 in Appendix D.2, on CelebA in Figure 6 in Appendix D.2, and on Tiny ImageNet in Figure 7 in Appendix D.2 as a qualitative comparison. Those generated images are consistent with the quantitative scores in Table 1.

**The HSW leads to faster convergence than the SW with a lower computation:** We plot the FID scores over training epochs of the SW and the HSW with the same setting as in Table 1 in Figure 3. We observe that FID scores from models trained by the HSW (with a better computation) reduce faster than ones trained from the SW. The same phenomenon happens with the IS scores in Figure 4 in Appendix D.2 where IS scores from the HSW increase earlier. The reason is that the HSW has a higher number of final projections than the SW, hence, it is a more discriminative signal than the SW.

**Ablation studies on $k$ and $L$ in the HSW:** In the Table 2, we report the additional FID scores, IS scores, the computational complexity ($\times 10^6$), and the projection complexity ($\times 10^6$) for the HSW with $(k, L) = \{(500, 1000), (500, 4000), (100, 50000)\}$. First, we see that given $k = 5000$, increasing $L$ from 1000 to 4000 improves the generative performance on all three datasets. Moreover, we observe that the HSW with the same number of final projections as the SW ($L = 1000$) gives comparable scores while the complexities are only about half of the complexities of the SW. When decreasing $k$ from 500 to 400 and increasing $L$ from 4000 to 6000, the generative performance of the HSW is enhanced and the complexities are also lighter. However, choosing a too-small $k$ and a too-large $L$ does not lead to a better result. For example, the HSW with $k = 100, L = 50000$ has high complexities compared to $k = 500, L = 4000$ and $k = 400, L = 6000$, however, its FID scores and IS scores are worse. The reason is because of the linearity of the HRT. In particular, the $k$ bottleneck projections form a subspace that has a rank at most $k$ and the $L$ final projections still lie in that subspace. This fact suggests that the value of $k$ should be also chosen to be sufficiently large compared to the true rank of the supports. In practice, data often lie on a low dimensional manifold, hence, $k$ can be chosen to be much smaller than $d$ and $L$ can be chosen to be large for a good estimation of discrepancy. This is an interpretable advantage of the HSW compared to the SW since it can separate between the assumption of the ranking of supports and the estimation of the discrepancy between measures.

**Max hierarchical sliced Wasserstein:** We also compare Max-HSW (see Definition 7 in Appendix B) with the conventional Max-SW in generative modeling in Table 3 in Appendix D.2. We observe that the overparameterization from HRT could also improve the optimization of finding good projections. We would like to recall that the Max-HSW is the generalization of the Max-SW, namely, Max-HSW with $k = 1$ is equivalent to the Max-SW. Therefore, the performance of the Max-HSW is at least the same as the Max-SW.

## 5 CONCLUSION

In this paper, we proposed a hierarchical approach to efficiently estimate the Wasserstein distance with provable benefits in terms of computation and memory. It formed final projections by combining linearly and randomly from a smaller set of bottleneck projections. We justified the main idea by introducing Hierarchical Radon Transform (HRT) and hierarchical sliced Wasserstein distance (HSW). We proved the injectivity of the HRT, the metricity of the HSW, and investigated its theoretical properties including computational complexity, sample complexity, projection complexity, and its connection to other sliced Wasserstein variants. Finally, we conducted experiments on deep generative modeling where the main computational burden due to the projection was highlighted. In this setting, HSW performed favorably in both generative quality and computational efficiency.

## ACKNOWLEDGEMENTS

NH acknowledges support from the NSF IFML 2019844 and the NSF AI Institute for Foundations of Machine Learning.

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

# Supplement to "Hierarchical Sliced Wasserstein Distance"

In this supplement, we first discuss some related works in Appendix A. We then present some additional materials including hierarchical Radon transform on multivariate Gaussian and mixture of multivariate Gaussians, max sliced Wasserstein distance, max hierarchical sliced Wasserstein (Max-HSW) distance, their computation, and their theoretical properties in Appendix B. After that, we collect proofs for key results in the paper in Appendix C. In Appendix D, we include the detailed training objectives of our generative modeling framework and some additional experiments including convergence in IS scores of generative models, generated images, and the comparison between the Max-HSW and the Max-SW. Moreover, we report experimental settings including parameter choices and neural network architectures in Appendix F.

## A    MORE RELATED WORKS

On the theoretical side, statistical properties of sliced Wasserstein distance in learning generative models are investigated in (Nadjahi et al., 2019). Distributional convergence of the Sliced Wasserstein process is derived in Xi & Niles-Weed (2022). Statistical, robustness, and computational guarantees for sliced Wasserstein distances are shown in (Nietert et al., 2022). A differential private version of the SW is proposed in (Rakotomamonjy & Liva, 2021).

On the methodological side, authors in (Nguyen et al., 2021a;b) replace uniform distribution on the projecting directions on the unit hyper-sphere in SW with an estimated distribution that puts high probabilities for discriminative directions. Spherical sliced Wasserstein which is a sliced Wasserstein variant on the hyper-sphere is introduced in (Bonet et al., 2022). Dependent projecting directions are utilized in (Nguyen et al., 2023b). Sliced partial optimal transport is proposed in (Bonneel & Coeurjolly, 2019; Bai et al., 2022) A fast biased approximation of the SW is proposed in (Nadjahi et al., 2021). The Augmented SW is introduced (Chen et al., 2022). A sliced Wasserstein variant between measures over tensors is defined in (Nguyen & Ho, 2022b). Estimating Wasserstein distance with one-dimensional transportation plans from orthogonal projecting directions is used in (Rowland et al., 2019). The SW is used in generative frameworks such as sliced iterative normalizing flows (Dai & Seljak, 2021) and fine-tuning pre-trained model (Lezama et al., 2021). The SW gradient flows are investigated in (Liutkus et al., 2019; Bonet et al., 2021). The SW is used for set representations in (Naderializadeh et al., 2021). Variational inference using the SW is carried out in (Yi & Liu, 2021). Similarly, the SW is used for approximate Bayesian computation in (Nadjahi et al., 2020).

## B    ADDITIONAL MATERIALS

**Hierarchical Radon Transform as the composition of Partial Radon Transform and Over-parametrized Radon Transform:** From the definitions of ORT (Definition 4) and PRT (Definition 3), we have:

$$
\begin{aligned}
(\mathcal{PR}(\mathcal{OR}f))(v, \theta_{1:k}, \psi) &= \int_{\mathbb{R}^k} \int_{\mathbb{R}^d} f(x) \prod_{i=1}^{k} \delta(t_i - \langle x, \theta_i \rangle) dx \delta\left(v - \sum_{i=1}^{k} t_i \psi_i\right) dt \\
&= \int_{\mathbb{R}^k} \int_{\mathbb{R}^d} f(x) \prod_{i=1}^{k} \delta(t_i - \langle x, \theta_i \rangle) \delta\left(v - \sum_{i=1}^{k} t_i \psi_i\right) dx dt \\
&= \int_{\mathbb{R}^d} \int_{\mathbb{R}^k} f(x) \prod_{i=1}^{k} \delta(t_i - \langle x, \theta_i \rangle) \delta\left(v - \sum_{i=1}^{k} t_i \psi_i\right) dt dx \\
&= \int_{\mathbb{R}^d} f(x) \delta\left(v - \sum_{i=1}^{k} \langle x, \theta_i \rangle \psi_i\right) dx \\
&:= (\mathcal{HR}f)(v, \theta_{1:k}, \psi),
\end{aligned}
$$

where the second and the third equality are due to the Fubini's theorem.

**Monte Carlo approximation error of HSW:** We have the following proposition for the Monte Carlo approximation error:

**Proposition 5.** *Given $p > 1$, $k > 1$, and $\mu, \nu \in \mathcal{P}_p(\mathbb{R}^d)$, we have the following error of Monte Carlo estimation for HSW:*

$$\mathbb{E}|\widehat{HSW}_{p,k}^p(\mu,\nu) - HSW_{p,k}^p(\mu,\nu)| \leq \frac{1}{\sqrt{HL}} Var\left[ W_p^p((\mathcal{HR}f_\mu)(\cdot, \theta_{1:k}, \psi), (\mathcal{HR}f_\nu)(\cdot, \theta_{1:k}, \psi)) \right]^{\frac{1}{2}}.$$

(7)

The proof is given in Appendix C.6. From the proposition, we see that the error reduces proportionally to the square root of $H$ and the square root of $L$. Therefore, increasing $H$ and $L$ could lead to better performance.

**Hierarchical Radon Transform of Multivariate Gaussian and Mixture of Multivariate Gaussians:** Similar to the result of Radon Transform for Multivariate Gaussian and Mixture of Multivariate Gaussians (Kolouri et al., 2018b), given projecting direction $\Theta = (\theta_{1:k}), \psi$, the one dimensional projected distributions from a Gaussian and a mixture of Gaussians with the HRT are also a Gaussian and a mixture of Gaussians respectively. These results are directly obtained from the linearity of Gaussian distributions. In more detail, let $f := \mathcal{N}(\mu, \Sigma)$, we have $(\mathcal{HR}f)(\cdot, \theta_{1:k}, \psi) = \mathcal{N}(\psi^\top \Theta^\top \mu, \psi^\top \Theta^\top \Sigma \Theta \psi)$. Similarly, let $f := \sum_{i=1}^k w_i \mathcal{N}(\mu_i, \Sigma_i)$, we have $(\mathcal{HR}f)(\cdot, \theta_{1:k}, \psi) = \sum_{i=1}^k w_i \mathcal{N}(\psi^\top \Theta^\top \mu_i, \psi^\top \Theta^\top \Sigma_i \Theta \psi)$.

**Max sliced Wasserstein distance:** We first review the definition of max sliced Wasserstein distance (Max-SW) (Deshpande et al., 2019).

**Definition 7.** *For any $p \geq 1$ and dimension $d \geq 1$, the max sliced Wasserstein-p distance between two probability measures $\mu \in \mathcal{P}_p(\mathbb{R}^d)$ and $\nu \in \mathcal{P}_p(\mathbb{R}^d)$ is given by:*

$$Max\text{-}SW_p(\mu,\nu) = \max_{\theta \in \mathbb{S}^{d-1}} W_p((\mathcal{R}f_\mu)(\cdot, \theta), (\mathcal{R}f_\nu)(\cdot, \theta)).$$

Max-SW is also a valid metric between probability measures (Deshpande et al., 2019). The benefit of Max-SW is that it uses only one projecting direction instead of $L$ projecting directions as SW. Therefore, its projection complexity is only $\mathcal{O}(d)$ compared to $\mathcal{O}(Ld)$ of SW. However, Max-SW requires an iterative optimization procedure to find the max projecting direction on the unit hypersphere e.g., projected gradient asscent (Kolouri et al., 2019b). We summarize the projected gradient ascent of the Max-SW in Algorithm 1.

---

**Algorithm 1** Max sliced Wasserstein distance

---

    **Input:** Probability measures: $\mu, \nu$, learning rate $\eta$, max number of iterations $T$.
    Initialize $\theta$
    **while** $\theta$ not converge or reach $T$ **do**
        $\theta = \theta + \eta \cdot \nabla_\theta W_p((\mathcal{R}f_\mu)(\cdot, \theta), (\mathcal{R}f_\nu)(\cdot, \theta))$
        $\theta = \frac{\theta}{||\theta||_2}$
    **end while**
    **Return:** $\theta, W_p((\mathcal{R}f_\mu)(\cdot, \theta), (\mathcal{R}f_\nu)(\cdot, \theta))$

---

**Max hierarchical sliced Wasserstein distance:** We now define the max hierarchical sliced Wasserstein distance.

**Definition 8.** *For any $p \geq 1$, $k \geq 1$ and dimension $d \geq 1$, the max hierarchical sliced Wasserstein-p distance between two probability measures $\mu \in \mathcal{P}_p(\mathbb{R}^d)$ and $\nu \in \mathcal{P}_p(\mathbb{R}^d)$ is given by:*

$$Max\text{-}HSW_{p,k}(\mu,\nu) = \max_{\theta_1,\ldots,\theta_k \in \mathbb{S}^{d-1}, \psi \in \mathbb{S}^{k-1}} W_p((\mathcal{HR}f_\mu)(\cdot, \theta_{1:k}, \psi), (\mathcal{HR}f_\nu)(\cdot, \theta_{1:k}, \psi)).$$

Similar to the idea of max sliced Wasserstein, Max-HSW improves the projection complexity of HSW by avoiding using Monte Carlo samples for projecting directions and mixing directions. In particular, Max-HSW finds the best set of bottleneck projections $\theta_{1:k}$ and the best mixing direction $\psi$ in terms of maximizing the discrepancy between two interested measures.

**Remark 1.** *We have that when $k = 1$, the max hierarchical sliced Wasserstein distance reverts into the max sliced Wasserstein distance.*

$$Max\text{-}HSW_{p,1}(\mu,\nu) = Max\text{-}SW_p(\mu,\nu),$$

*This is due to the fact that $\psi \in \mathbb{S}^0 = \{1\}$.*

In Max-HSW, we do not enforce any linear independence constraint over $k$ bottleneck projection directions. However, that kind of constraint can be done by forcing the $k$ vectors to be orthogonal. Formally, we could project $k$ vectors into the Stiefel manifold by performing the Gram-Schmidt process. We can only do maximization over $k$ bottleneck vectors and take the expectation over the linear mixing vectors. We denote this variant as Semi Max hierarchical sliced Wasserstein distance.

**Definition 9.** *For any $p \geq 1$, $k \geq 1$ and dimension $d \geq 1$, the max hierarchical sliced Wasserstein-p distance between two probability measures $\mu \in \mathcal{P}_p(\mathbb{R}^d)$ and $\nu \in \mathcal{P}_p(\mathbb{R}^d)$ is given by:*

$$\text{SemiMax-HSW}_{p,k}(\mu,\nu) = \max_{\theta_1,\ldots,\theta_k \in \mathbb{S}^{d-1}} \left( \mathbb{E}_{\psi \sim \mathcal{U}(\mathbb{S}^{k-1})} W_p^p \left( (\mathcal{HR}f_\mu)(\cdot,\theta_{1:k},\psi), (\mathcal{HR}f_\nu)(\cdot,\theta_{1:k},\psi) \right) \right)^{\frac{1}{p}}.$$

The optimization can be done by stochastic gradient ascent that utilizes Monte Carlo samples $\psi_1,\ldots,\psi_L$ from $\mathcal{U}(\mathbb{S}^{k-1})$.

**Computation of max hierarchical sliced Wasserstein distance:** Similar to the Max-SW, Max-HSW can be computed via the projected gradient ascent algorithm. In practice, the gradient of $\theta_{1:k}$ and $\psi$ can be computed by using backpropagation (chain rule) since the hierarchical Radon Transform can be seen as a two-layer neural network. We summarize the projected gradient ascent of the Max-HSW in Algorithm 2.

---

**Algorithm 2** Max hierarchical sliced Wasserstein distance

---

    **Input:** Probability measures: $\mu, \nu$, learning rate $\eta$, max number of iterations $T$.
    Initialize $\theta$
    **while** $\theta_{1:k}, \psi$ not converge or reach $T$ **do**
        $\psi = \psi + \eta \cdot \nabla_\psi W_p \left( (\mathcal{HR}f_\mu)(\cdot,\theta_{1:k},\psi), (\mathcal{HR}f_\nu)(\cdot,\theta_{1:k},\psi) \right)$
        $\psi = \frac{\psi}{||\psi||_2}$
        **for** $i = 1$ to $k$ **do**
            $\theta_i = \theta_i + \eta \cdot \nabla_{\theta_i} W_p \left( (\mathcal{HR}f_\mu)(\cdot,\theta_{1:k},\psi), (\mathcal{HR}f_\nu)(\cdot,\theta_{1:k},\psi) \right)$
            $\theta_i = \frac{\theta_i}{||\theta_i||_2}$
        **end for**
    **end while**
    **Return:** $\theta_{1:k}, \psi, W_p \left( (\mathcal{HR}f_\mu)(\cdot,\theta_{1:k},\psi), (\mathcal{HR}f_\nu)(\cdot,\theta_{1:k},\psi) \right)$

---

**Properties of max hierarchical sliced Wasserstein distance:** We first have the following result for the metricity of Max-HSW.

**Theorem 2.** *For any $p \geq 1$ and $k \geq 1$, the hierarchical sliced Wasserstein Max-HSW$_{p,k}(\cdot,\cdot)$ is a metric on the space of probability measures on $\mathbb{R}^d$.*

Proof of Theorem 2 is given in Appendix C.7. We establish the connection between the HSW, max hierarchical sliced Wasserstein (Max-HSW), max sliced Wasserstein (Max-SW), and Wasserstein distance in Proposition 3. The proof is given in Appendix C.4.

We demonstrate that the max hierarchical sliced Wasserstein does not suffer from the curse of dimensionality for the inference purpose, namely, the sample complexity for the empirical distribution from i.i.d. samples to approximate their underlying distribution is at the order of $\mathcal{O}(n^{-1/2})$.

**Proposition 6.** *Assume that $P$ is a probability measure that has supports on compact set of $\mathbb{R}^d$. Let $X_1, X_2, \ldots, X_n$ be i.i.d. samples from $P$ and we denote $P_n = \frac{1}{n} \sum_{i=1}^n \delta_{X_i}$ as the empirical measure on data samples. Then, for any $p \geq 1$ and $k \geq 1$, there exists a universal constant $C > 0$ such that*

$$\mathbb{E}[\text{Max-HSW}_{p,k}(P_n, P)] \leq Ck\sqrt{(d+1)\log n/n},$$

*where the outer expectation is taken with respect to the data $X_1, X_2, \ldots, X_n$.*

Proof of Proposition 6 is similar to the proof of Proposition 4. We refer the reader to Appendix C.5.

**Hierarchical Generalized Sliced Wasserstein distance:** We first define the Overparameterized Generalized Radon Transform, Partial Generalized Radon Transform, and Hierarchical Generalized Radon Transform. We recall two injective defining functions in (Kolouri et al., 2019b) which are the circular function $g(x,\theta) = \|x - r * \theta\|_2$ for $x \in \mathbb{R}^d, \theta \in \mathbb{S}^{d-1}$, (hyper-parameter $r \in \mathbb{R}^+$) and the

homogeneous polynomials with an odd degree $g(x, \theta) = \sum_{|\alpha|=m} \theta_\alpha x^\alpha$ where $\alpha = (\alpha_1, \ldots, \alpha_{d_\alpha}) \in \mathbb{N}^{d_\alpha}, |\alpha| = \sum_{i=1}^{d_\alpha} \alpha_i, x^\alpha = \prod_{i=1}^{d_\alpha} x_i^{\alpha_i}$. The summation in the polynomial defining function iterates over all possible multi-indices $\alpha$ such that $|\alpha| = m$ with $m$ denotes the degree of the polynomial and $\theta \in \mathbb{S}^{d_\alpha-1}$.

**Definition 10** (Partial Generalized Radon Transform). *Given the defining function $g(x, \theta)$ : $\mathbb{R}^{d_1} \times \Omega_\theta \to \mathbb{R}$, the Partial Generalized Radon Transform (PGRT) $\mathcal{PR} : \mathbb{L}_1(\mathbb{R}^{d_1} \times \mathbb{R}^{d_2}) \to \mathbb{L}_1(\mathbb{R} \times \Omega_\theta \times \mathbb{R}^{d_2})$ is defined as:*

$$(\mathcal{PGR}f)(t, \theta, y) = \int_{\mathbb{R}^{d_1}} f(x, y)\delta(t - g(x, \theta))dx. \tag{8}$$

**Definition 11** (Overparameterized Generalized Radon Transform). *Given the defining function $g(x, \theta) : \mathbb{R}^d \times \Omega_\theta \to \mathbb{R}$, the Overparameterized Radon Transform (OGRT) $\mathcal{OR} : \mathbb{L}_1(\mathbb{R}^d) \to \mathbb{L}_1(\mathbb{R}^{\otimes k} \times \mathbb{S}^{(d-1)\otimes k})$ is defined as:*

$$(\mathcal{OGR}f)(t_{1:k}, \theta_{1:k}) = \int_{\mathbb{R}^d} f(x) \prod_{i=1}^k \delta(t_i - g(x, \theta_i))dx, \tag{9}$$

*where $t_{1:k} := (t_1, \ldots, t_k) \in \mathbb{R}^{\otimes k}$ and $\theta_{1:k} := (\theta_1, \ldots, \theta_k) \in (\mathbb{S}^{d-1})^{\otimes k}$.*

**Definition 12** (Hierarchical Generalized Radon Transform). *Given the defining functions $g_1(x, \theta)$ : $\mathbb{R}^d \times \Omega_\theta \to \mathbb{R}$ and $g_2(x, \psi) : \mathbb{R}^k \times \Omega_\psi \to \mathbb{R}$, Hierarchical Generalized Radon Transform (HGRT) $\mathcal{HGR} : \mathbb{L}_1(\mathbb{R}^d) \to \mathbb{L}_1(\mathbb{R} \times \Omega_\theta^{\otimes k} \times \Omega_\psi)$ is defined as:*

$$(\mathcal{HGR}f)(v, \theta_{1:k}, \psi) = \int_{\mathbb{R}^d} f(x)\delta(v - g_2((g_1(x, \theta_1), \ldots, g_1(x, \theta_k)), \psi))\, dx, \tag{10}$$

*where $v \in \mathbb{R}, \psi \in \Omega_\psi$ and $\theta_{1:k} = (\theta_1, \ldots, \theta_k) \in \Omega_\theta^{\otimes k}$.*

The injectivity of PGRT, OGRT, and HGRT follow the injectivity of GRT, i.e., the proof technique is similar to the case of ORT and HRT. We refer the reader to the previous work (Kolouri et al., 2019b) for more information. The benefit of HGRT is that it can create a much stronger non-linearity by stacking multiple injective ones.

**Definition 13.** *For any $p \geq 1$, $k \geq 1$, and dimension $d \geq 1$, the hierarchical generalized sliced Wasserstein distance of order $p$ between two probability measures $\mu \in \mathcal{P}_p(\mathbb{R}^d)$ and $\nu \in \mathcal{P}_p(\mathbb{R}^d)$ is given by:*

$$HGSW_{p,k}(\mu, \nu) = \left( \mathbb{E}_{\theta_{1:k}, \psi} W_p^p((\mathcal{HGR}f_\mu)(\cdot, \theta_{1:k}, \psi), (\mathcal{HGR}f_\nu)(\cdot, \theta_{1:k}, \psi)) \right)^{\frac{1}{p}}, \tag{11}$$

*where $\theta_1, \ldots, \theta_k \sim \mathcal{U}(\Omega_\theta)$ and $\psi \sim \mathcal{U}(\Omega_\psi)$. The proof technique for the metricity of HGSW is the same as HSW which is based on the metricity of the Wasserstein distance, the injectivity of HGRT, and the Minkowski inequality.*

**Distributional Hierarchical Sliced Wasserstein distance:** In HSW, projecting directions are drawn from the uniform distribution which might not be the best choice in some settings. We could replace it with some other distributions, e.g., Von Mises Fisher (Nguyen et al., 2021b) or implicit distribution (Nguyen et al., 2021a).

**Definition 14.** *For any $p \geq 1$, $k \geq 1$, and dimension $d \geq 1$, the distributional hierarchical sliced Wasserstein distance of order $p$ between two probability measures $\mu \in \mathcal{P}_p(\mathbb{R}^d)$ and $\nu \in \mathcal{P}_p(\mathbb{R}^d)$ is given by:*

$$DHSW_{p,k}(\mu, \nu)$$
$$= \sup_{\sigma_1 \in \Gamma(\mathbb{S}^{d-1}), \sigma_2 \in \Gamma(\mathbb{S}^{k-1})} \left( \mathbb{E}_{\theta_{1:k} \sim \sigma_1^{\otimes k}, \psi \sim \sigma_2} W_p^p((\mathcal{HR}f_\mu)(\cdot, \theta_{1:k}, \psi), (\mathcal{HR}f_\nu)(\cdot, \theta_{1:k}, \psi)) \right)^{\frac{1}{p}}, \tag{12}$$

*where $\Gamma(\mathbb{S}^{d-1})$ ($\Gamma(\mathbb{S}^{k-1})$) is a family of distributions over the unit-hypersphere. The metricity of DHSW requires the continuity condition of $\Gamma(\mathbb{S}^{d-1})$ and $\Gamma(\mathbb{S}^{k-1})$, i.e., they have supports on all the unit-hypersphere to guarantee the injectivity of the HRT. We refer the reader to previous works (Nguyen et al., 2021a;b) for more details.*

**Augmented approaches for HSW:** Authors in (Chen et al., 2022) propose to augment measures to higher dimensions for better linear separation. We extend the framework to our hierarchical transform approach. We first define the Spatial Hierarchical Radon transform.

**Definition 15** (Spatial Hierarchical Radon Transform). *Given a injective mapping $g : \mathbb{R}^d \to \mathbb{R}^{d'}$ $(d' \geq d)$, Spatial Hierarchical Radon Transform (SHRT) $\mathcal{SHR} : \mathbb{L}_1(\mathbb{R}^d) \to \mathbb{L}_1\left(\mathbb{R} \times \mathbb{S}^{(d'-1)\otimes k} \times \mathbb{S}^{k-1}\right)$ is defined as:*

$$(\mathcal{SHR}f)(v, \theta_{1:k}, \psi) = \int_{\mathbb{R}^d} f(x) \delta\left(v - \sum_{i=1}^{k} \langle g(x), \theta_i \rangle \psi_i\right) dx, \tag{13}$$

*where $v \in \mathbb{R}, \psi = (\psi_1, \dots, \psi_k) \in \mathbb{S}^{k-1}$, and $\theta_{1:k} = (\theta_1, \dots, \theta_k) \in (\mathbb{S}^{d'-1})^{\otimes k}$.*

We can further extend the SHRT to hierarchical spatial Radon Transform (HSRT) by using the augmenting mappings twice.

**Definition 16** (Hierarchical Spatial Radon Transform). *Given injective mapping $g_1 : \mathbb{R}^d \to \mathbb{R}^{d'}$ $(d' \geq d)$ and $g_2 : \mathbb{R}^k \to \mathbb{R}^{k'}$ $(k' \geq k)$, Hierarchical Spatial Radon Transform (SHRT) $\mathcal{SHR} : \mathbb{L}_1(\mathbb{R}^d) \to \mathbb{L}_1\left(\mathbb{R} \times \mathbb{S}^{(d'-1)\otimes k} \times \mathbb{S}^{k'-1}\right)$ is defined as:*

$$(\mathcal{HSR}f)(v, \theta_{1:k}, \psi) = \int_{\mathbb{R}^d} f(x) \delta\left(v - \langle g_2((\langle g_1(x), \theta_1 \rangle, \dots, \langle g_1(x), \theta_k \rangle)\psi\rangle\right) dx, \tag{14}$$

*where $v \in \mathbb{R}, \psi = (\psi_1, \dots, \psi_k) \in \mathbb{S}^{k'-1}$, and $\theta_{1:k} = (\theta_1, \dots, \theta_k) \in (\mathbb{S}^{d'-1})^{\otimes k}$.*

From the above two transforms, we could derive the Augmented Hierarchical sliced Wasserstein (AHSW) distance and Hierarchical Augmented sliced Wasserstein (HASW) distance respectively. We skip the definitions of AHSW and HASW here since they are straightforward from the definition of HSW. The injectivity of SHRT and HSRT and the metricity of augmented distances can be proven based on the injectivity of augmenting mappings $g_1$ and $g_2$. We refer the reader to the previous work of Chen et al. (2022) for more details.

## C  PROOFS

In this appendix, we provide proofs for key results in the main text and in Appendix B.

### C.1  PROOF OF PROPOSITION 1

Let us consider functions $f, g \in \mathbb{L}^1(\mathbb{R}^d)$ such that
$$(\mathcal{OR}f)(t_{1:k}, \theta_{1:k}) = (\mathcal{OR}g)(t_{1:k}, \theta_{1:k}),$$
It is clear from the Definition 4 that
$$\int_{\mathbb{R}^d} f(x) \prod_{i=1}^{k} \delta(t_i - \langle x, \theta_i \rangle) dx = \int_{\mathbb{R}^d} g(x) \prod_{i=1}^{k} \delta(t_i - \langle x, \theta_i \rangle) dx.$$
Taking the integral of both sides in the above equation with respect to $(k-1)$ variables $t_2, \dots, t_k$, we get
$$\int_{\mathbb{R}^{\otimes(k-1)}} \int_{\mathbb{R}^d} f(x) \prod_{i=1}^{k} \delta(t_i - \langle x, \theta_i \rangle) dx dt_2 \dots dt_k = \int_{\mathbb{R}^{\otimes(k-1)}} \int_{\mathbb{R}^d} g(x) \prod_{i=1}^{k} \delta(t_i - \langle x, \theta_i \rangle) dx dt_2 \dots dt_k.$$
Next, by applying the Fubini's theorem, we have
$$\int_{\mathbb{R}^d} f(x) \delta(t_1 - \langle x, \theta_1 \rangle) \left(\prod_{i=2}^{k} \int_{\mathbb{R}} \delta(t_i - \langle x, \theta_i \rangle) dt_i\right) dx = \int_{\mathbb{R}^d} g(x) \delta(t_1 - \langle x, \theta_1 \rangle) \left(\prod_{i=2}^{k} \int_{\mathbb{R}} \delta(t_i - \langle x, \theta_i \rangle) dt_i\right) dx.$$
Note that for any $i = 2, \dots, k-1$, by using change of variables $s_i = t_i - \langle x, \theta_i \rangle$, we have $\int_{\mathbb{R}} \delta(t_i - \langle x, \theta_i \rangle) dt_i = \int_{\mathbb{R}} \delta(s_i) ds_i = 1$. As a result,
$$\int_{\mathbb{R}^d} f(x) \delta(t_1 - \langle x, \theta_1 \rangle) dx = \int_{\mathbb{R}^d} f(x) \delta(t_1 - \langle x, \theta_1 \rangle) dx,$$
or equivalently, $(\mathcal{R})f(t_1, \theta_1) = \mathcal{R}g(t_1, \theta_1)$. Recall that the Radon transform $\mathcal{R}$ is injective. Thus, we obtain that $f = g$, which completes the proof.

## C.2  Proof of Proposition 2

Let us consider arbitrary functions $f, g \in \mathbb{L}^1(\mathbb{R}^d)$ satisfying

$$(\mathcal{HR}f)(v, \theta_{1:k}, \psi) = (\mathcal{HR}g)(v, \theta_{1:k}, \psi),$$

where $v \in \mathbb{R}, \psi \in \mathbb{S}^{k-1}$ and $\theta_{1:k} \in (\mathbb{S}^{d-1})^{\otimes k}$. It can be seen from the beginning of Appendix B that $(\mathcal{HR}f)(v, \theta_{1:k}, \psi) = (\mathcal{PR}(\mathcal{OR}f))(v, \psi, \theta_{1:k})$ and $(\mathcal{HR}g)(v, \theta_{1:k}, \psi) = (\mathcal{PR}(\mathcal{OR}g))(v, \psi, \theta_{1:k})$. Therefore, we get

$$(\mathcal{PR}(\mathcal{OR}f))(v, \psi, \theta_{1:k}) = (\mathcal{PR}(\mathcal{OR}g))(v, \psi, \theta_{1:k}).$$

Since Partial Radon Transform is injective, we obtain $h_f = h_g$, or equivalently,

$$(\mathcal{OR}f)(t_{1:k}, \theta_{1:k}) = (\mathcal{OR}g)(t_{1:k}, \theta_{1:k}),$$

which leads to $f = g$ due to the injectivity of Overparametrized Radon Transform. Hence, we reach the conclusion of the proposition.

## C.3  Proof of Theorem 1

To prove that the hierarchical sliced Wasserstein $HSW_{p,k}(\cdot, \cdot)$ is a metric on the space of all probability measure on $\mathbb{R}^d$ for any $p, k \geq 1$, we need to verify four following criteria:

**Symmetry:** For any $p, k \geq 1$, it is obvious that $HSW_{p,k}(\mu, \nu) = HSW_{p,k}(\nu, \mu)$ for any probability measures $\mu$ and $\nu$.

**Non-negativity:** The non-negativity of $HSW_{p,k}(\cdot, \cdot)$ comes directly from the non-negativity of the Wasserstein metric.

**Identity:** For any $p, k \geq 1$, it is clear that when $\mu = \nu$, we have $HSW_{p,k}(\mu, \nu) = 0$. Now, assume that $HSW_{p,k}(\mu, \nu) = 0$, then $W_p((\mathcal{HR}f_\mu)(\cdot, \theta_{1:k}, \psi), (\mathcal{HR}f_\nu)(\cdot, \theta_{1:k}, \psi)) = 0$ for almost all $\psi \in \mathbb{S}^{k-1}, \theta_{1:k} \in (\mathbb{S}^{d-1})^{\otimes k}$. By applying the identity property of the Wasserstein distance, we have $(\mathcal{HR}f_\mu)(\cdot, \theta_{1:k}, \psi) = (\mathcal{HR}f_\nu)(\cdot, \theta_{1:k}, \psi)$ for almost all $\psi \in \mathbb{S}^{k-1}, \theta_{1:k} \in (\mathbb{S}^{d-1})^{\otimes k}$. Since the Hierarchical Radon Transform is injective, we obtain $f_\mu = f_\nu$, which implies that $\mu = \nu$.

**Triangle Inequality:** For any probability measures $\mu_1, \mu_2, \mu_3$, we find that

$$
\begin{aligned}
HSW_{p,k}(\mu_1, \mu_3) &= \left(\mathbb{E}_{\theta_{1:k}, \psi} W_p^p\left((\mathcal{HR}f_{\mu_1})(\cdot, \theta_{1:k}, \psi), (\mathcal{HR}f_{\mu_3})(\cdot, \theta_{1:k}, \psi)\right)\right)^{\frac{1}{p}} \\
&\leq \left(\mathbb{E}_{\theta_{1:k}, \psi}[W_p^p\left((\mathcal{HR}f_{\mu_1})(\cdot, \theta_{1:k}, \psi), (\mathcal{HR}f_{\mu_2})(\cdot, \theta_{1:k}, \psi)\right) \right. \\
&\quad \left. + W_p^p\left((\mathcal{HR}f_{\mu_2})(\cdot, \theta_{1:k}, \psi), (\mathcal{HR}f_{\mu_3})(\cdot, \theta_{1:k}, \psi)\right)]\right)^{\frac{1}{p}} \\
&\leq \left(\mathbb{E}_{\theta_{1:k}, \psi} W_p^p\left((\mathcal{HR}f_{\mu_1})(\cdot, \theta_{1:k}, \psi), (\mathcal{HR}f_{\mu_2})(\cdot, \theta_{1:k}, \psi)\right)\right)^{\frac{1}{p}} \\
&\quad + \left(\mathbb{E}_{\theta_{1:k}, \psi} W_p^p\left((\mathcal{HR}f_{\mu_2})(\cdot, \theta_{1:k}, \psi), (\mathcal{HR}f_{\mu_3})(\cdot, \theta_{1:k}, \psi)\right)\right)^{\frac{1}{p}} \\
&= HSW_{p,k}(\mu_1, \mu_2) + HSW_{p,k}(\mu_2, \mu_3),
\end{aligned}
$$

where the first inequality is due to the triangle inequality of Wasserstein metric, namely, we have

$$
\begin{aligned}
W_p\left((\mathcal{HR}f_{\mu_1})(\cdot, \theta_{1:k}, \psi), (\mathcal{HR}f_{\mu_3})(\cdot, \theta_{1:k}, \psi)\right) &\leq W_p\left((\mathcal{HR}f_{\mu_1})(\cdot, \theta_{1:k}, \psi), (\mathcal{HR}f_{\mu_2})(\cdot, \theta_{1:k}, \psi)\right) \\
&\quad + W_p\left((\mathcal{HR}f_{\mu_2})(\cdot, \theta_{1:k}, \psi), (\mathcal{HR}f_{\mu_3})(\cdot, \theta_{1:k}, \psi)\right),
\end{aligned}
$$

while the second inequality is an application of the Minkowski inequality for integrals.

Hence, the hierarchical sliced Wasserstein $HSW_{p,k}(\cdot, \cdot)$ is a metric on the space of all probability measures on $\mathbb{R}^d$ for any $p, k \geq 1$.

## C.4  Proof of Proposition 3

The proof of this proposition is direct from the definition of the hierarchical sliced Wasserstein distance, the sliced Wasserstein distance, the max hierarchical sliced Wasserstein distance, and the max sliced Wasserstein distance. Here, we provide the proof for the completeness.

(a) We start with

$$
\begin{aligned}
HSW_{p,k}(\mu,\nu) &= \left(\mathbb{E}_{\theta_{1:k},\psi} W_p^p\left((\mathcal{H}\mathcal{R}f_\mu)(\cdot,\theta_{1:k},\psi),(\mathcal{H}\mathcal{R}f_\nu)(\cdot,\theta_{1:k},\psi)\right)\right)^{\frac{1}{p}} \\
&\leq \max_{\theta_1,\ldots,\theta_k\in\mathbb{S}^{d-1},\psi\in\mathbb{S}^{k-1}} W_p\left((\mathcal{H}\mathcal{R}f_\mu)(\cdot,\theta_{1:k},\psi),(\mathcal{H}\mathcal{R}f_\nu)(\cdot,\theta_{1:k},\psi)\right), \\
&:= \text{Max-HSW}_{p,k}(\mu,\nu).
\end{aligned}
$$

Subsequently, we have

$$
\begin{aligned}
\text{Max-HSW}_{p,k}(\mu,\nu) &= \max_{\theta_1,\ldots,\theta_k\in\mathbb{S}^{d-1},\psi\in\mathbb{S}^{k-1}} W_p\left((\mathcal{H}\mathcal{R}f_\mu)(\cdot,\theta_{1:k},\psi),(\mathcal{H}\mathcal{R}f_\nu)(\cdot,\theta_1,\ldots,\theta_k,\psi)\right) \\
&= \max_{\theta_1,\ldots,\theta_k\in\mathbb{S}^{d-1},\psi\in\mathbb{S}^{k-1}} \left(\inf_{\pi\in\Pi(\mu,\nu)}\int_{\mathbb{R}^d\times\mathbb{R}^d}\left|\psi^\top\Theta^\top x-\psi^\top\Theta^\top y\right|^p d\pi(x,y)\right)^{\frac{1}{p}} \\
&= \max_{\theta_1,\ldots,\theta_k\in\mathbb{S}^{d-1},\psi\in\mathbb{S}^{k-1}} \left(\inf_{\pi\in\Pi(\mu,\nu)}\int_{\mathbb{R}^d\times\mathbb{R}^d}\left|\sum_{j=1}^k\psi_j\theta_j^\top(x-y)\right|^p d\pi(x,y)\right)^{\frac{1}{p}} \\
&\leq \max_{\theta_1,\ldots,\theta_k\in\mathbb{S}^{d-1},\psi\in\mathbb{S}^{k-1}} \left(\inf_{\pi\in\Pi(\mu,\nu)}\int_{\mathbb{R}^d\times\mathbb{R}^d}||\psi||^p\left|\sum_{j=1}^k\theta_j^\top(x-y)\right|^p d\pi(x,y)\right)^{\frac{1}{p}} \\
&= \max_{\theta_1,\ldots,\theta_k\in\mathbb{S}^{d-1}} \left(\inf_{\pi\in\Pi(\mu,\nu)}\int_{\mathbb{R}^d\times\mathbb{R}^d}\left|\sum_{j=1}^k\theta_j^\top(x-y)\right|^p d\pi(x,y)\right)^{\frac{1}{p}} \\
&\leq \max_{\theta\in\mathbb{S}^{d-1}} \left(\inf_{\pi\in\Pi(\mu,\nu)}\int_{\mathbb{R}^d\times\mathbb{R}^d}k^p\left|\theta^\top x-\theta^\top y\right|^p d\pi(x,y)\right)^{\frac{1}{p}} \\
&= k\cdot\max_{\theta\in\mathbb{S}^{d-1}} \left(\inf_{\pi\in\Pi(\mu,\nu)}\int_{\mathbb{R}^d\times\mathbb{R}^d}\left|\theta^\top x-\theta^\top y\right|^p d\pi(x,y)\right)^{\frac{1}{p}} \\
&= k\cdot\max_{\theta\in\mathbb{S}^{d-1}} W_p\left(\theta\sharp\mu,\theta\sharp\nu\right) \\
&= k\cdot\max_{\theta\in\mathbb{S}^{d-1}} W_p\left((\mathcal{R})f_\mu(\cdot,\theta),(\mathcal{R})f_\nu(\cdot,\theta)\right) \\
&= k\cdot\text{Max-SW}_p(\mu,\nu).
\end{aligned}
$$

Finally, by applying the Cauchy-Schwartz inequality, we get

$$
\begin{aligned}
\text{Max-SW}_p^p(\mu,\nu) &= \max_{\theta\in\mathbb{S}^{d-1}} \left(\inf_{\pi\in\Pi(\mu,\nu)}\int_{\mathbb{R}^d}|\theta^\top x-\theta^\top y|^p d\pi(x,y)\right) \\
&\leq \max_{\theta\in\mathbb{S}^{d-1}} \left(\inf_{\pi\in\Pi(\mu,\nu)}\int_{\mathbb{R}^d\times\mathbb{R}^d}\|\theta\|^p\|x-y\|^p d\pi(x,y)\right) \\
&= \inf_{\pi\in\Pi(\mu,\nu)}\int_{\mathbb{R}^d\times\mathbb{R}^d}\|\theta\|^p\|x-y\|^p d\pi(x,y) \\
&= W_p^p(\mu,\nu).
\end{aligned}
$$

Putting the above results together, we obtain the conclusion of the proposition.

(b) Firstly, it is obvious that

$$
SW_p(\mu,\nu) = \left(\mathbb{E}_{\theta\sim\mathcal{U}(\mathbb{S}^{d-1})}W_p^p\left((\mathcal{R}f_\mu)(\cdot,\theta),(\mathcal{R}f_\nu)(\cdot,\theta)\right)\right)^{\frac{1}{p}} \leq \max_{\theta\in\mathbb{S}^{d-1}} W_p\left((\mathcal{R}f_\mu)(\cdot,\theta),(\mathcal{R}f_\nu)(\cdot,\theta)\right).
$$

Subsequently, let $\theta_1' = \theta^* = \arg\max_{\theta \in \mathbb{S}^{d-1}} W_p\left((\mathcal{R}f_\mu)(\cdot, \theta), (\mathcal{R}f_\nu)(\cdot, \theta)\right)$ and $\psi' = (1, 0, \ldots, 0) \in \mathbb{S}^{k-1}$. Then, for any $\theta_1' \in \mathbb{S}^{d-1}$, we have

$$
\begin{aligned}
\text{Max-HSW}_{p,k}(\mu, \nu) &= \max_{\theta_1, \ldots, \theta_k \in \mathbb{S}^{d-1}, \psi \in \mathbb{S}^{k-1}} W_p\left((\mathcal{H}\mathcal{R}f_\mu)(\cdot, \theta_{1:k}, \psi), (\mathcal{H}\mathcal{R}f_\nu)(\cdot, \theta_{1:k}, \psi)\right) \\
&\geq W_p\left((\mathcal{H}\mathcal{R}f_\mu)(\cdot, \theta_1', \theta_2, \ldots, \theta_k, \psi'), (\mathcal{H}\mathcal{R}f_\nu)(\cdot, \theta_1', \theta_2, \ldots, \theta_k, \psi')\right) \\
&= \left(\inf_{\pi \in \Pi(\mu, \nu)} \int_{\mathbb{R}^d} |\theta_1^\top x - \theta_1^\top y|^p \, d\pi(x, y)\right)^{\frac{1}{p}} \\
&= W_p\left((\mathcal{R}f_\mu)(\cdot, \theta_1'), (\mathcal{R}f_\nu)(\cdot, \theta_1')\right).
\end{aligned}
$$

Since the above result holds for all arbitrary $\theta_1' \in \mathbb{S}^{d-1}$, we obtain

$$
\text{Max-HSW}_{p,k}(\mu, \nu) \geq \max_{\theta \in \mathbb{S}^{d-1}} W_p\left((\mathcal{R}f_\mu)(\cdot, \theta), (\mathcal{R}f_\nu)(\cdot, \theta)\right) = \text{Max-SW}_p(\mu, \nu),
$$

which completes the proof.

## C.5 PROOF OF PROPOSITION 4

For the ease of the presentation, we denote $\Theta \subset \mathbb{R}^d$ as the compact set of the probability measure $P$. The result of Proposition 3 indicates that we have

$$
\mathbb{E}[HSW_{p,k}(P_n, P)] \leq \mathbb{E}\left[k \cdot \text{Max-SW}_p(P_n, P)\right],
$$

where we define

$$
\text{Max-SW}_p(P_n, P) := \max_{\theta \in \mathbb{S}^{d-1}} W_p((\mathcal{R}f_{P_n})(\cdot, \theta), (\mathcal{R}f_P)(\cdot, \theta)) := \max_{\theta \in \mathbb{S}^{d-1}} W_p(\theta \sharp P_n, \theta \sharp P).
$$

Therefore, the conclusion of the proposition follows as long as we can demonstrate that

$$
\mathbb{E}[\text{Max-SW}_p(P_n, P)] \leq C\sqrt{(d+1)\log_2 n / n}
$$

where $C > 0$ is some universal constant and the outer expectation is taken with respect to the data. We first start with the property of the closed-form expression of Wasserstein distance in one dimension, which leads to the following equations:

$$
\begin{aligned}
\text{Max-SW}_p^p(P_n, P) &= \max_{\theta \in \mathbb{S}^{d-1}} \int_0^1 |F_{n,\theta}^{-1}(u) - F_\theta^{-1}(u)|^p du \\
&= \max_{\theta \in \mathbb{R}^d : \|\theta\| = 1} \int_0^1 |F_{n,\theta}^{-1}(u) - F_\theta^{-1}(u)|^p du \\
&\leq \max_{\theta \in \mathbb{R}^d : \|\theta\| \leq 1} \int_{\mathbb{R}} |F_{n,\theta}(x) - F_\theta(x)|^p dx \\
&\leq \text{diam}(\Theta) \max_{\theta \in \mathbb{R}^d : \|\theta\| \leq 1} |F_{n,\theta}(x) - F_\theta(x)|^p,
\end{aligned}
$$

where we denote $F_{n,\theta}$ and $F_\theta$ as the cumulative distributions of the two push-forward distributions $\theta \sharp P_n$ and $\theta \sharp P$.

Direct calculation indicates that

$$
\max_{\theta \in \mathbb{R}^d : \|\theta\| \leq 1} |F_{n,\theta}(x) - F_\theta(x)| = \sup_{B \in \mathcal{B}} |P_n(B) - P(B)|,
$$

where we denote $\mathcal{B}$ as the set of half-spaces $\{z \in \mathbb{R}^d : \theta^\top z \leq x\}$ for all $\theta \in \mathbb{R}^d$ such that $\|\theta\| \leq 1$. From Wainwright (2019), it is known that the Vapnik-Chervonenkis (VC) dimension of $\mathcal{B}$ is at most $d + 1$. Therefore, arrive at

$$
\sup_{B \in \mathcal{B}} |P_n(B) - P(B)| \leq \sqrt{\frac{32}{n}[(d+1)\log_2(n+1) + \log_2(8/\delta)]}
$$

with probability at least $1 - \delta$. Collecting the above results, we finally obtain that

$$
\mathbb{E}[\text{Max-SW}_p(P_n, P)] \leq C\sqrt{(d+1)\log_2 n / n},
$$

where $C > 0$ is some universal constant. As a consequence, the conclusion of the proposition follows.

## C.6    PROOF OF PROPOSITION 5

Using the Holder's inequality, we have:

$$
\mathbb{E}|\widehat{HSW}_{p,k}^{p}(\mu, \nu) - HSW_{p,k}^{p}(\mu, \nu)|
$$

$$
\leq \left( \mathbb{E}|\widehat{HSW}_{p,k}^{p}(\mu, \nu) - HSW_{p,k}^{p}(\mu, \nu)|^{2} \right)^{\frac{1}{2}}
$$

$$
= \left( \mathbb{E} \left| \frac{1}{HL} \sum_{h=1}^{H} \sum_{l=1}^{L} W_p^p \left( (\mathcal{HR}f_\mu)(\cdot, \theta_{1:k,h}, \psi_l), (\mathcal{HR}f_\nu)(\cdot, \theta_{1:k,h}, \psi_l) \right) \right. \right.
$$

$$
\left. \left. - \mathbb{E}_{\theta_{1:k},\psi} W_p^p \left( (\mathcal{HR}f_\mu)(\cdot, \theta_{1:k}, \psi), (\mathcal{HR}f_\nu)(\cdot, \theta_{1:k}, \psi) \right) \right|^{2} \right)^{\frac{1}{2}}
$$

$$
= \left( Var \left[ \frac{1}{HL} \sum_{h=1}^{H} \sum_{l=1}^{L} W_p^p \left( (\mathcal{HR}f_\mu)(\cdot, \theta_{1:k,h}, \psi_l), (\mathcal{HR}f_\nu)(\cdot, \theta_{1:k,h}, \psi_l) \right) \right] \right)^{\frac{1}{2}}
$$

$$
= \frac{1}{\sqrt{HL}} Var \left[ W_p^p \left( (\mathcal{HR}f_\mu)(\cdot, \theta_{1:k}, \psi), (\mathcal{HR}f_\nu)(\cdot, \theta_{1:k}, \psi) \right) \right]^{\frac{1}{2}},
$$

which completes the proof.

## C.7    PROOF OF THEOREM 2

**Symmetry:** For any $p \geq 1$, it is clear that Max-HSW$_p(\mu, \nu) = $ Max-HSW$_p(\nu, \mu)$ for any probability measures $\mu$ and $\nu$.

**Non-negativity:** The non-negativity of Max-HSW$_{p,k}(\cdot, \cdot)$ comes directly from the non-negativity of the Wasserstein metric.

**Existence of the max directions:** The unit hyperspheres $\mathbb{S}^{d-1}$ and $\mathbb{S}^{k-1}$ are compact and we have $W_p \left( (\mathcal{HR}f_\mu)(\cdot, \theta_{1:k}, \psi), (\mathcal{HR}f_\nu)(\cdot, \theta_{1:k}, \psi) \right)$ is continuous in terms of $\theta_{1:k}$ and $\psi$. Therefore, there exists

$$
\theta_{1:k}^*, \psi^* = \operatorname*{arg\,max}_{\theta_1,\ldots,\theta_k \in \mathbb{S}^{d-1}, \psi \in \mathbb{S}^{k-1}} W_p \left( (\mathcal{HR}f_\mu)(\cdot, \theta_{1:k}, \psi), (\mathcal{HR}f_\nu)(\cdot, \theta_{1:k}, \psi) \right)
$$

**Identity:** For any $p \geq 1$ and $k \geq 1$, it is clear that when $\mu = \nu$, then Max-HSW$_p(\mu, \nu) = 0$. When Max-HSW$_p(\mu, \nu) = 0$, we have $W_p((\mathcal{HR}f_\mu)(\cdot, \theta_{1:k}, \psi), (\mathcal{HR}f_\nu)(\cdot, \theta_{1:k}, \psi)) = 0$ for almost all $\psi \in \mathbb{S}^{k-1}$, $\theta_{1:k} \in (\mathbb{S}^{d-1})^{\otimes k}$. Applying the identity property of the Wasserstein distance, we have $(\mathcal{HR}f_\mu)(\cdot, \theta_{1:k}, \psi) = (\mathcal{HR}f_\nu)(\cdot, \theta_{1:k}, \psi)$ almost all $\psi \in \mathbb{S}^{k-1}$, $\theta_{1:k} \in (\mathbb{S}^{d-1})^{\otimes k}$. Since the Hierarchical Radon Transform is injective, we obtain $\mu = \nu$.

**Triangle Inequality:** For any probability measures $\mu_1, \mu_2, \mu_3$, we find that

$$
\begin{aligned}
\text{Max-HSW}_{p,k}(\mu_1, \mu_3) &= \max_{\theta_1,\ldots,\theta_k \in \mathbb{S}^{d-1}, \psi \in \mathbb{S}^{k-1}} W_p \left( (\mathcal{HR}f_{\mu_1})(\cdot, \theta_{1:k}, \psi), (\mathcal{HR}f_{\mu_3})(\cdot, \theta_{1:k}, \psi) \right) \\
&= W_p \left( (\mathcal{HR}f_{\mu_1})(\cdot, \theta_{1:k}^*, \psi^*), (\mathcal{HR}f_{\mu_3})(\cdot, \theta_{1:k}^*, \psi^*) \right) \\
&\leq W_p \left( (\mathcal{HR}f_{\mu_1})(\cdot, \theta_{1:k}^*, \psi^*), (\mathcal{HR}f_{\mu_2})(\cdot, \theta_{1:k}^*, \psi^*) \right) \\
&\quad + W_p \left( (\mathcal{HR}f_{\mu_2})(\cdot, \theta_{1:k}^*, \psi^*), (\mathcal{HR}f_{\mu_3})(\cdot, \theta_{1:k}^*, \psi^*) \right) \\
&\leq \max_{\theta_1,\ldots,\theta_k \in \mathbb{S}^{d-1}, \psi \in \mathbb{S}^{k-1}} W_p \left( (\mathcal{HR}f_{\mu_1})(\cdot, \theta_{1:k}, \psi), (\mathcal{HR}f_{\mu_2})(\cdot, \theta_{1:k}, \psi) \right) \\
&\quad + \max_{\theta_1,\ldots,\theta_k \in \mathbb{S}^{d-1}, \psi \in \mathbb{S}^{k-1}} W_p \left( (\mathcal{HR}f_{\mu_2})(\cdot, \theta_{1:k}, \psi), (\mathcal{HR}f_{\mu_3})(\cdot, \theta_{1:k}, \psi) \right) \\
&= \text{Max-HSW}_{p,k}(\mu_1, \mu_2) + \text{Max-HSW}_{p,k}(\mu_2, \mu_3)
\end{aligned}
$$

where the first equality is due to the existence of the max directions, the first inequality is due to the triangle inequality with Wasserstein metric, and the second inequality is an application of the definition of maxima.

# D DEEP GENERATIVE MODELS

In this section, we first discuss the way we train generative models in our experiments with sliced Wasserstein variants. We then provide additional experimental results including convergence speed of the SW and the HSW in IS score, generated images from the SW and the HSW, and comparison between the Max-SW and the Max-HSW.

## D.1 TRAINING DETAIL

In short, we follow the generative framework from Deshpande et al. (2018). The model distribution is parameterized as $\mu_\phi(x) \in \mathcal{P}(\mathbb{R}^d)$ where $\mu_\phi(x) = \mathcal{G}_\phi \sharp \mu_0$, $\mu_0 := \mathcal{N}(0, I_{128})$, and $\mathcal{G}_\phi$ is a Resnet He et al. (2016)-type neural network. The detail of the neural networks is given in Appendix F. Previous works suggest to use a discriminator as a type of ground metric learning since the true metric between images is not given (unknown). We denote the discriminator as a composite function $D_{\beta_2} \circ D_{\beta_1}$ where $D_{\beta_1} : \mathbb{R}^d \to \mathbb{R}^{d'}$ and $D_{\beta_2} : \mathbb{R}^{d'} \to \mathbb{R}$. In particular, $D_{\beta_1}$ transforms original images to their latent features. After that, $D_{\beta_2}$ maps their features maps to their corresponding discriminative scores. We denote the data distribution is $\nu_{data}$, our objectives are:

$$\min_\phi \mathbb{E}_{X \sim \nu_{data}^{\otimes m}, Y \sim \mu_0^{\otimes m}} \mathcal{D}(D_{\beta_1} \sharp P_X, D_{\beta_1} \sharp \mathcal{G}_\phi \sharp P_Y),$$

$$\min_{\beta_1, \beta_2} \left( \mathbb{E}_{x \sim \nu_{data}}[\min(0, -1 + D_{\beta_2}(D_{\beta_1}(x)))] + \mathbb{E}_{z \sim \mu_0}[\min(0, -1 - D_{\beta_2}(D_{\beta_1}(\mathcal{G}_\phi(z))))] \right),$$

where $m \geq 1$ is the mini-batch size and $\mathcal{D}(\cdot, \cdot)$ is the (Max-)SW distance or (Max-)HSW distance.

**Relation to $m$-mini-batch energy distance:** The objective for training the generator seen as a type of $m$-mini-batch energy distance Klebanov et al. (2005); Salimans et al. (2018) with sliced Wasserstein variants kernel. Moreover, it is also known as mini-batch optimal transport Fatras et al. (2020); Nguyen et al. (2022a;b).

## D.2 ADDITIONAL RESULTS

**Convergence of generative models from SW and HSW in IS scores:** We plot the IS scores over training epochs of the SW and the HSW with the same setting as in Table 1 in Figure 4. We observe that IS scores from models trained by the HSW (with better computation) increase faster than ones trained from the SW. We would like to recall that reason is that the HSW has a higher number of final projections than the SW, hence, it has a more discriminative signal than the SW.

**Radom generated images:** We show randomly generated images from CIFAR10, CelebA, and Tiny Imagenet from the SW and the HSW generative models in Figure 5- 7. From these images, we observe that increasing $L$ enhance the performance of the SW, and increasing both $L$ and $k$ yield better images for the HSW. Also, the qualitative comparison from generated images between the SW and the HSW is consistent with the FID scores and IS scores in Table 1.

**Comparison between the Max-HSW and the Max-SW :** We would like to recall that Max-HSW is the generalization of the Max-SW since Max-HSW with $k = 1$ is equivalent to the Max-SW. Therefore, the performance of Max-HSW is a least the same as the Max-SW. To find out the benefit of $k > 1$ in Max-HSW, we run Max-HSW with $k \in \{1, 10, 100, 1000)$, the slice learning rate $\eta \in \{0.001, 0.01, 0.1\}$, and the maximum number of iteration $T \in \{1, 10, 100\}$. We report the best FID scores and IS scores in Table 3. We observe that Max-HSW $k > 1$ gives a better FID score than the Max-SW on CelebA dataset. The reason might be due to the overparametrization of the Max-HSW that leads to a better final slice. However, on CIFAR10 and Tiny ImageNet, the Max-HSW $k > 1$ does not show any improvement compared to the Max-HSW $k = 1$ (Max-SW). This might be because the projected gradient ascent does not work well in a multiplayer structure like in HRT. We will leave the investigation about optimization of the Max-HSW in future works since the Max-HSW has the potential to explain partially adversarial training frameworks.

# E COLOR TRANSFER VIA GRADIENT FLOW

In the color transfer setting, we are interested in transferring the color palette of a source image to the color palette of a target image. Formally, we denote the color palette of the source image

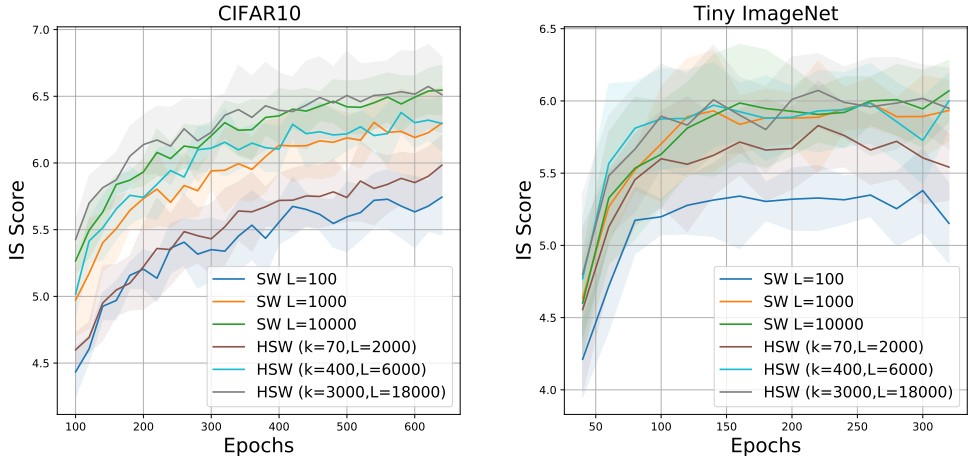

Figure 4: The IS scores over epochs of different training losses on datasets. We observe that HSW helps the generative models converge faster.

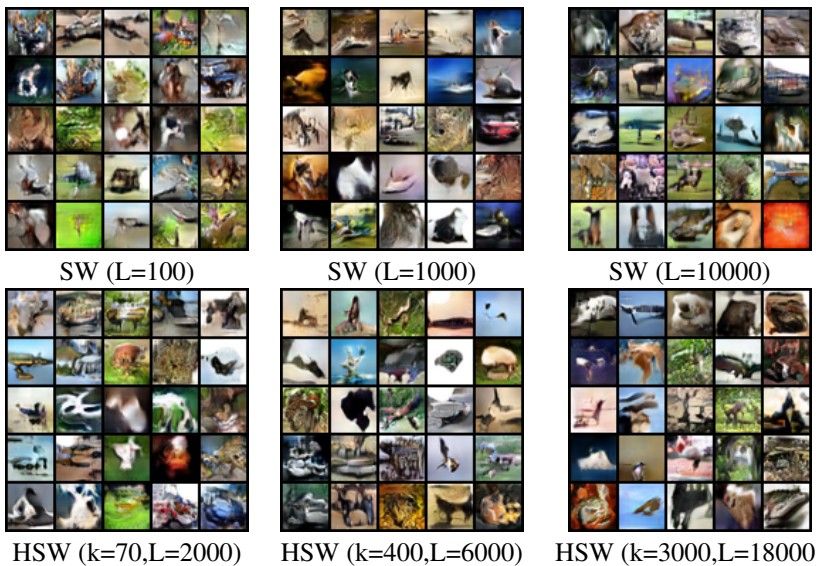

Figure 5: Random generated images of the SW and the HSW on CIFAR10.

Table 3: FID scores and IS scores the Max-SW and the Max-HSW of on CIFAR10 (32x32), CelebA (64x64), and Tiny ImageNet (64x64).

| Method | CIFAR10 | | CelebA | Tiny ImageNet | |
|---|---|---|---|---|---|
| | FID ($\downarrow$) | IS ($\uparrow$) | FID ($\downarrow$) | FID ($\downarrow$) | IS ($\uparrow$) |
| Max-SW (Max-HSW k=1) | 43.67±2.34 | 6.49±0.11 | 17.17±1.72 | 82.47±5.73 | 6.03±0.52 |
| Max-HSW | 43.67±2.34 | 6.49±0.11 | **15.92±0.87** | 82.47±5.73 | 6.03±0.52 |

and the target image as $X = (x_1, \ldots, x_n)$ and $Y = (y_1, \ldots, y_n)$ of size $n \times 3$ (RGB) in turn with $n > 1$. We would like to note that, the orders of points in $X$ and $Y$ are corresponding to the pixels in images. We denote $P_X = \frac{1}{n} \sum_{i=1}^{n} \delta_{x_i}$ and $P_Y = \frac{1}{n} \sum_{i=1}^{n} \delta_{y_i}$ as two empirical measures on $X$ and $Y$ respectively. We now move $P_X$ to $P_Y$ under the following gradient flow: $\dot{X}(t) = \nabla_X D(P_X, P_Y)$ with $D$ as a chosen distance, e.g., SW and HSW. We simply use the Euler scheme to solve this gradient flow with $T$ iterations. Since color is represented by three values in the set $\{0, \ldots, 255\}$, we round values in $X(T)$ to the closest values in $\{0, \ldots, 255\}$ to obtain the final color palette.

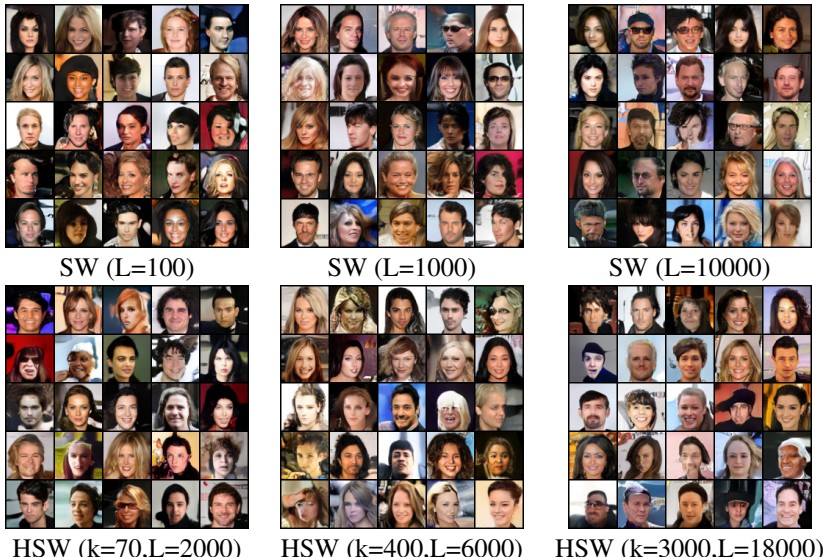

Figure 6: Random generated images of the SW and the HSW on CelebA.

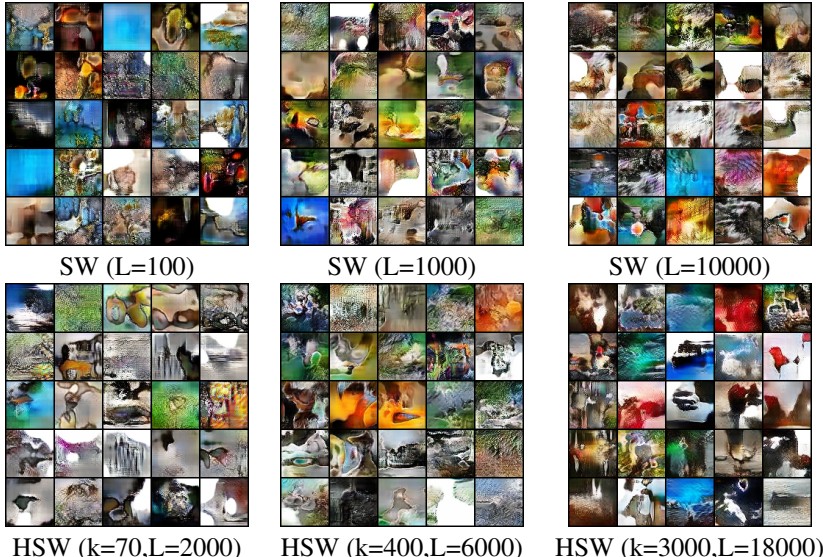

Figure 7: Random generated images of the SW and the HSW on Tiny ImageNet.

In our experiments, we compress original images to $3000$ colors by using K-Means before doing color transfer. In this case, $n = 3000$ is much larger than $d = 3$, hence, using HSW does not give any computational benefits compared to SW. Namely, computing one-dimension Wasserstein distances becomes the main computation. However, HSW could provide the dependency between final projection directions since they are created from the same set of bottleneck projection directions.

In Figure 8, we compare SW, generalized sliced Wasserstein (GSW) (Kolouri et al., 2019b), HSW, and hierarchical generalized sliced Wasserstein (HGSW) which is a straightforward extension of HSW by replacing the Radon Transform variants by Generalized Radon Transform variants. All the distances are computed with the order 2 ($p = 2$) and we use the number of iterations $T = 10000$ with the step size 0.1 for the Euler scheme. For GSW and HGSW, we use the polynomial of degree 3 as the defining function. For SW and GSW, we set the number of projections $L = 3$. For HSW and HGSW, we set the number of bottleneck projections $k = 10$ and the number of final projections $L = 3$. In the figure, we report the computational time and the Wasserstein-2 distance between the corresponding color palette and the color palette of the target image. We observe that HSW gives better Wasserstein-2 scores than SW in both two shown images while the computation time is comparable. Similarly, the same

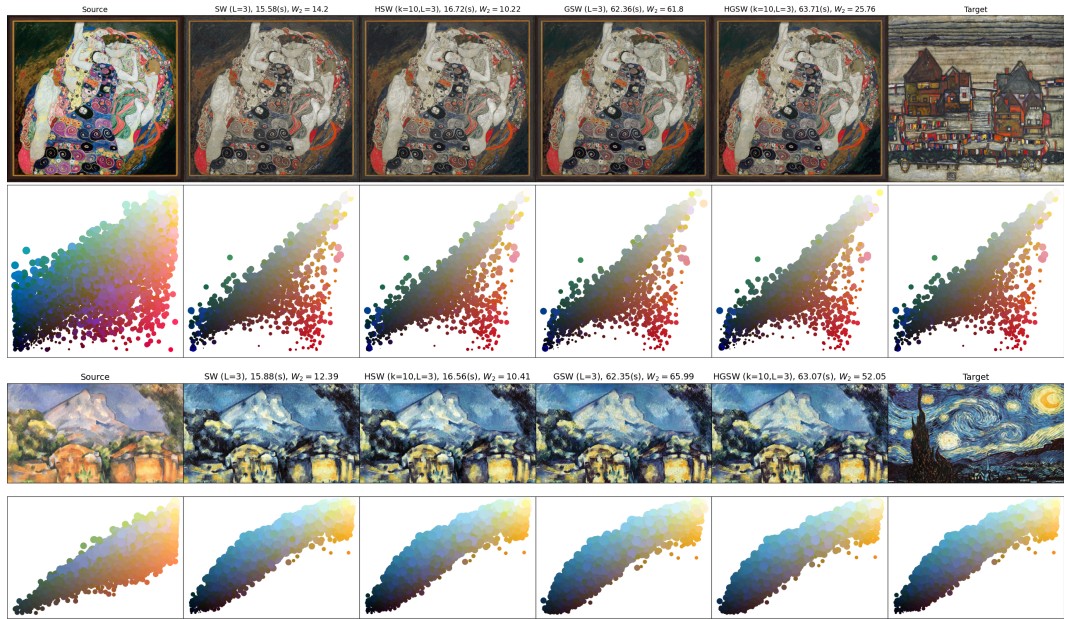

Figure 8: Color transfered images and color palette with SW, HSW, GSW, and HGSW gradient flows. The computational time and Wasserstein-2 scores between corresponding color palate and the color palette of the target images are given next to the name of the distances.

phenomenon happens in the case of GSW and HGSW. Comparing non-linear projecting and linear projecting, non-linear variants cost about four times more computation while their Wasserstein-2 scores are higher. However, it is worth noting that a better Wasserstein-2 score is not necessarily better since a non-linear transform creates a different flavor. Overall, the hierarchical approach in HSW and HGSW induces different gradient flows compared to SW and GSW which may yield some potential benefits.

## F    ADDITIONAL EXPERIMENTAL SETTINGS

**Additional settings:**  For all datasets, the number of training iterations is set to 50000. We update the generator $\mathcal{G}_\phi$ for each 5 iterations using (Max-)SW and (Max-)HSW. Moreover, we update $D_{\beta_1}$ and $D_{\beta_2}$ ( the discriminator) every iterations. The mini-batch size $m$ is set 128 in all datasets. We set the learning rate for $\mathcal{G}_\phi$, $D_{\beta_1}$, and $D_{\beta_2}$ to 0.0002. The optimizer that we use is Adam (Kingma & Ba, 2014) with parameters $(\beta_1, \beta_2) = (0, 0.9)$ (slightly abuse of notations). For SW and HSW, we use $p = 2$ (the order of Wasserstein distance).

**FID scores and IS scores:**  We use 50000 random samples from trained models for computing the FID scores and the Inception scores. In evaluating FID scores, we use all training samples for computing statistics of datasets[3].

**Neural networks:** We present the architectures of our generative networks and the discriminative networks on CIFAR10, CelebA, and Tiny ImageNet as follow:.

- **CIFAR10**:
    - $\mathcal{G}_\phi$: $\epsilon \in \mathbb{R}^{128}(\sim \mathcal{N}(0,1)) \to 4 \times 4 \times 256$(Dense,linear) $\to$ ResBlock up 256 $\to$ ResBlock up 256 $\to$ ResBlock up 256 $\to$ BN, ReLU, $\to 3 \times 3$ conv, 3 Tanh .
    - $D_{\beta_1}$: $\boldsymbol{x} \in [-1,1]^{32 \times 32 \times 3} \to$ ResBlock down 128 $\to$ ResBlock down 128 $\to$ ResBlock down 128 $\to$ ResBlock 128 $\to$ ResBlock 128.
    - $D_{\beta_2}$: $\boldsymbol{x} \in \mathbb{R}^{128 \times 8 \times 8} \to$ ReLU $\to$ Global sum pooling(128) $\to$ 1(Spectral normalization).

---

[3]We use the scores calculation from https://github.com/GongXinyuu/sngan.pytorch.

- **CelebA and Tiny ImageNet:**
  - $\mathcal{G}_\phi$: $\boldsymbol{\epsilon} \in \mathbb{R}^{128}(\sim \mathcal{N}(0,1)) \rightarrow 4 \times 4 \times 256(\text{Dense,linear}) \rightarrow$ ResBlock up $256 \rightarrow$ ResBlock up $256 \rightarrow$ ResBlock up $256 \rightarrow$ ResBlock up $256 \rightarrow$ BN, ReLU, $\rightarrow$ $3 \times 3$ conv, 3 Tanh .
  - $D_{\beta_1}$: $\boldsymbol{x} \in [-1,1]^{32 \times 32 \times 3} \rightarrow$ ResBlock down $128 \rightarrow$ ResBlock down $128 \rightarrow$ ResBlock down $128 \rightarrow$ ResBlock $128 \rightarrow$ ResBlock $128$.
  - $D_{\beta_2}$: $\boldsymbol{x} \in \mathbb{R}^{128 \times 8 \times 8} \rightarrow$ ReLU $\rightarrow$ Global sum pooling$(128) \rightarrow$ $1(\text{Spectral normalization})$.

