# OpenReview forum: "Hierarchical Sliced Wasserstein Distance"
_ICLR.cc/2023/Conference — ICLR 2023 poster_

### Official Review · Reviewer_YGxc · 2022-10-14

**Confidence:** 3
**Correctness:** 4
**Technical Novelty And Significance:** 4
**Empirical Novelty And Significance:** 3
**Recommendation:** 6

**Clarity, Quality, Novelty And Reproducibility:**

The paper is well-written, of good quality and the information provided in the paper should be sufficient to reproduce the experiments, I believe.

**Strength And Weaknesses:**

* Strengths
    * The method is sounded.
    * The paper is well motivated. Notably, the introduction clearly states the contribution of the paper, and is easy to read despite the technicity of the proposition
* Weaknesses
    * The paper is already very dense (_eg._ parts of the experimental results discussed in the paper are only available as supplementary material) yet it raises several questions that would be worth investigating in the paper.

In more details, I really enjoyed reading the paper and believe the contribution is interesting.
However, I have the following remarks / questions:
* The paper is already dense, and I get that one has to make choices, but I believe an application of HRT on other sliced divergences would have helped illustrate the genericity of the proposition (though the matter is discussed in the paragraph "Applications of HRT").
* Do we have any intuition for a link between HSW and SW (apart from the links exhibited in Proposition 3 between HSW and Max-SW -- resp. SW and Max-HSW)?
    * When reading Proposition 3, it is unclear to me why Max-HSW would differ from Max-SW since, in spirit, they both correspond to taking the max over projections on a line. This is probably due to a misunderstanding on my side, but maybe adding a discussion on this point could help the reader better grasp the difference between those two.
    * This is also related to the discussion on "Distributions of final projecting directions in HRT", which concludes:
        > To our knowledge, the manifold $\mathcal{S}$ has not been explored sufficiently in previous works, which may be a potential direction of future research"
        * I believe providing informed thoughts on properties of this manifold would be a real plus for this paper.
* The section on the benefits of HSW when $d \gg n$ is not very convincing since all complexities are said to be "proportional to XXX". Either the constant is the same for all and this should be stated clearly or a quantitative study of the running times as a function of $L$ would be welcome.
    * This is also reflected in the experiments: could the "computational complexity" criterion be replaced by running times to make sure that in Table 1, for example (but this holds for other results too), each sub-group of methods indeed corresponds to cases where the HSW computation is cheaper and nothing is hidden in the $\mathcal{O}(\cdot)$ notations. The same applies to the "Projection complexity" criterion.


I have a few extra minor comments:
* "which has columns are" -> "which columns are"
* [Titouan et al., 2019] -> [Vayer et al., 2019]
* End of Section 3: the last sentence ends weirdly, probably a bad copy-paste
* In Figure 3, using markers could help identify paired methods (which HSW variant corresponds, in terms of computation, to which SW variant)
* "In the Table 2, we reports" (p. 8) -> report

**Summary Of The Paper:**

This paper introduces the Hierarchical Sliced Wasserstein (HSW) distance that is a variant over the Sliced Wasserstein (SW) distance.
In a nutshell, the HSW distance improves on the computational complexity of SW in cases where $d \gg n$ by lowering the burden related to the projections.
To do so, projections on subspaces of dimension $k$ is performed and the final $L$ projections are projections from this $k$-dimensional space to the real line.
Both properties of this HSW and experimental validation on generative modeling tasks are presented.

**Summary Of The Review:**

The proposed method is very interesting and the paper is well-written.
I would appreciate more details in places (see previous section) and a comparison of complexity based on running times to make sure the comparison is fair.

---

> ### Author Response · Authors · 2022-11-12
> **Response to Reviewer YGxc**
>
> Thank you for your time and your feedback. We have revised the paper based on your suggestions. All modifications are written in blue color.
>
> **Q13**: The paper is already dense, and I get that one has to make choices, but I believe an application of HRT on other sliced divergences would have helped illustrate the genericity of the proposition (though the matter is discussed in the paragraph "Applications of HRT").
>
> **A13**: Radon Transform has played an important role in defining sliced probability divergences [2]. Similar to the sliced Wasserstein distance that uses the Wasserstein distance as the “base” divergence between projected measures, we could define other sliced probability divergences by replacing the Wasserstein distance with other divergences e.g., Sinkhorn divergence [2], Cramer distance [3], Fisher divergence [4], and so on. By replacing Radon Transform with Hierarchical Radon Transform,  we obtain hierarchical sliced probability divergences. For each base divergence, the computational complexity is changed and the nature of the divergence is also changed. Therefore, it leads to a lot of effort to investigate both the theoretical and empirical properties of those divergences. In the paper, we focus on the sliced Wasserstein distance since it is the most famous sliced probability divergence that has both computational benefits (near-linear) and statistical benefits (no curse of dimensionality).
>
> [2] Statistical and Topological Properties of Sliced Probability Divergences, Nadjahi et al.
>
> [3] Sliced Cramer Synaptic Consolidation for Preserving Deeply Learned Representations, Kolouri et al.
>
> [4] Sliced Score Matching: A Scalable Approach to Density and Score Estimation, Song et al.
>
> **Q14**: Do we have any intuition for a link between HSW and SW (apart from the links exhibited in Proposition 3 between HSW and Max-SW -- resp. SW and Max-HSW)?
>
> **A14**: When $k=1$, HSW will revert into SW. Also, if we use a discrete distribution which supports are standard basis vectors in $k$ dimension for the linear mixing vector $\psi$ instead of the uniform distribution over the unit-hypersphere, HSW will become SW. However, in the general case, the link between HSW and SW is not trivial. Hence, we will leave this investigation to future work.
>
> **Q15**: When reading Proposition 3, it is unclear to me why Max-HSW would differ from Max-SW since, in spirit, they both correspond to taking the max over projections on a line. This is probably due to a misunderstanding on my side, but maybe adding a discussion on this point could help the reader better grasp the difference between those two.
>
> **A15**: The set of final projecting directions is not the same as the unit-hypersphere, hence, Max-HSW and Max-SW are two different optimization problems. Also, Max-HSW has more parameters than Max-SW which might lead to a better optimization landscape like in deep learning.
>
> **Q16**: Properties of this manifold $\mathcal{S}$ would be a real plus for this paper.
>
> **A16**:  The only thing that we can tell about the manifold $\mathcal{S}^{d-1}$ is that it contains the unit-hypersphere $\mathbb{S}^{d-1}$. In particular, if $\psi$ is a vector that belongs to the set of standard basis vectors, e.g, $\psi =(1,0,\ldots,0)$, then $\psi^\top \Theta = \theta_1 \in \mathcal{S}^{d-1}$.
>
> **Q17**: Could the "computational complexity" criterion be replaced by running times?
>
> **A17**: We have added the running times for SW and HSW in Table 1 in the revision. We observe that they are consistent with the computational complexity analysis. Namely, the computation in each sub-group of methods is also the same. For memory, we observe that all settings show almost the same memory consumption since the main computation and memory in deep generative modeling are consumed by neural networks. In the first version, the reason that we use computational complexity and projection complexity analysis is that the running time and memory depend on the status of the computational hardware which is hard to control.
>
> **Q18**: Typos
>
> **A18**: We have fixed all the typos based on your suggestion in our revision.

---

> > ### Comment · Reviewer_YGxc · 2022-11-15
> > **Response to the authors**
> >
> > Thank you for this detailed answer.
> >
> > Regarding Q15, I am sorry to say that your answer does not help me grasp why the final projecting directions are not the same for both Max-HSW and Max-SW since in Proposition 3 you are taking the max over all possible projections. Could you exhibit a direction that is in the unit hypershpere and cannot be realized by the 2-step projection in HSW, or the reverse? Or could you at least point to a bibliographical reference that motivates your claim. Sorry if this is straight-forward, but your answer is not sufficient for me to understand the difference between Max-SW and Max-HSW.
> >
> > Overall, given your answer, I will stick to my initial rating.
> >
> > Best regards

---

> > > ### Author Response · Authors · 2022-11-17
> > > **Response to Reviewer YGxc**
> > >
> > > Dear Reviewer YGxc,
> > >
> > > Thank you for your response. To ease the ensuing discussion, we first recall the definitions of Max-SW and Max-HSW in the push-forward measure form. Namely,
> > >
> > > $\text{Max-SW}(\mu,\nu;p) = \max_{\theta \in \mathbb{S}^{d-1}} \left( \inf_{\pi \in \Pi(\mu, \nu)} \int_{\mathbb{R}^d \times \mathbb{R}^d}\left|\theta^\top x-\theta^\top y\right|^p d \pi(x, y) \right)^{\frac{1}{p}}, $
> > >
> > > $\text{Max-HSW}(\mu,\nu;p) = \max_{\theta_1, \ldots, \theta_k \in \mathbb{S}^{d-1}, \psi \in \mathbb{S}^{k-1}} \left( \inf_{\pi \in \Pi(\mu, \nu)} \int_{\mathbb{R}^d \times \mathbb{R}^d}\left|\psi^\top \Theta^\top x- \psi^\top \Theta^\top y\right|^p d \pi(x, y) \right)^{\frac{1}{p}},$
> > >
> > > where $\Theta =(\theta_1,\ldots,\theta_k)$ is the matrix of size $d\times k$.
> > >
> > > Assume that $\theta_{1}^*, \ldots, \theta_{k}^*, \psi^*$ are the optimal solutions of the Max-HSW and $\theta^*$ is the optimal solution of the Max-SW. We denotes $\theta’ =  \psi^* \Theta^* = \sum_{i = 1}^{k} \psi_{i}^* \theta_{i}^*$ where $\psi^* = (\psi_{1}^*, \ldots, \psi_{k}^*)$. Since we do not have exact closed-form expressions for $\theta_{1}^*, \ldots, \theta_{k}^*, \psi^*$ (note that, the simulations may only yield the local solutions), we consider two cases of these optimal solutions:
> > >
> > > Case 1: $\theta_{1}^*, \ldots, \theta_{k}^*$ are not identical, i.e., at least two of them are different. Then, since they belong to the unit sphere $\mathbb{S}^{d-1}$ and that set is not convex, it indicates that $\theta’$, a linear combination of $\theta_{1}^*, \ldots, \theta_{k}^*$, does not necessarily belong to $\mathbb{S}^{d-1}$. Therefore, under this setting, $\theta’$ is not necessarily identical to $\theta^*$, which implies that the value of Max-HSW can be different from Max-SW.
> > >
> > > Case 2: $\theta_{1}^*, \ldots, \theta_{k}^*$ are identical, then $ \theta’ = (\sum_{i = 1}^{k} \psi_{i}^*) \theta_{1}^* $. It indicates that $	\lVert \theta' 	\rVert_{2} =  |\sum_{i = 1}^{k} \psi_{i}^* | $ . If we would like $ \theta’  \in \mathbb{S}^{d-1} $, we need to have $ | \sum_{i = 1}^{k} \psi_{i}^*| = 1  $. Therefore, unless the optimal $\psi^* \in \mathbb{S}^{k - 1} $, i.e., $ \sum_{i = 1}^{k} (\psi_{i}^*)^2 = 1$, satisfies $|\sum_{i = 1}^{k} \psi_{i}^{*}| = 1$, the value of Max-HSW can be different from Max-SW.
> > >
> > > We would like to emphasize that since we do not know the exact values of $ \theta_{1}^*, \ldots, \theta_{k}^*, \psi^* $ and the simulations only yield local minima values for approximating $\theta_{1}^*, \ldots, \theta_{k}^*, \psi^*$, it is in general hard to check Case 2 and condition $|\sum_{i = 1}^{k} \psi_{i}^*| = 1$. Given that it is not the main focus of the paper, which is mainly about the HSW, we leave the careful investigation of whether Case 2 will hold for the future work.

---

> > > > ### Comment · Reviewer_YGxc · 2022-11-17
> > > > **Thank you for this clear explanation**
> > > >
> > > > OK, that's clearer now, and I agree that this does not have to be discussed in the paper. Thank you very much.
> > > >
> > > > Overall, this discussion confirms my positive opinion on the paper.

---

### Official Review · Reviewer_EUsv · 2022-10-24

**Confidence:** 5
**Correctness:** 4
**Technical Novelty And Significance:** 3
**Empirical Novelty And Significance:** Not applicable
**Recommendation:** 8

**Clarity, Quality, Novelty And Reproducibility:**

* The paper is well-written, and the concepts are crystal clear.

* The paper is of high quality.

* The proposed concept and the resulting algorithm are very simple and practical. The Overparametrized Radon and the Hierarchical Radon transforms are novel and could be of interest to the community.

**Strength And Weaknesses:**

### Strengths:

* The paper is well-written and well-motivated.
* The paper is theoretically sound, and the experiments support the authors' claims.
* While conceptually simple and algorithmically trivial to implement, the approach is well supported by the theory of the proposed Hierarchical Radon transform, rooting the simple algorithm into the solid mathematical ground.
* The ablation study in Table 2 is really great!

### Weaknesses:

I do not see any major flaws in this paper. However, below are some of the points that, in my opinion, could have improved the paper:

* When defining the Max-HSW variation, the maximization could be performed on the $k$ bottleneck slices (I find this very interesting), on the $L$ linear combinations of the bottleneck slices, or jointly (as it is currently defined in the paper). It would have been exciting to see the effect of only maximizing over the $k$ bottleneck slices. Moreover, it seems that one should require the $k$ bottleneck slicers to be linearly independent; what enforces this constraint?

* I am not sure if I follow these sentences: "Since HGSW has more than one non-linear transform layer, it has the provision of using a more complex non-linear transform than the conventional Generalized sliced Wasserstein.  Compared to the neural network defining function of GRT, HGSW preserves the metricity." As far as I can see, if any of the projections (i.e., the $k$ bottleneck or the $L$ following slices) become nonlinear, then the metricity of HGSW would also depend on the defining functions and the injectivity of the (in this case) generalized hierarchical Radon transform. Could you please clarify this?

* Currently Propositions 1 and 2 could read as if $HRf$ and $HRg$ are equal for **some** $\theta_{1:k}$ and $\psi$ then $f=g$, which I am sure we agree that is not true. I think what the proposition is saying is if $HRf(\cdot,\theta_{1:k},\psi)=HRg(\cdot,\theta_{1:k},\psi)$ for $\forall \theta_{1:k}, \psi$ then $f=g$. Maybe rewording the propositions could increase the clarity.

### Minor editorial:

Typo - Page 3: "the set of absolutely integrable function on" should be "the set of absolutely integrable functions on"


**Summary Of The Paper:**

The paper proposes an extension of the Sliced Wasserstein (SW) distance, denoted as Hierarchical Sliced Wasserstein (HSW) distance. At its core, for $d$ dimensional samples and $L$ slices, HSW first chooses $k$ bottleneck projections and then applies $L$ linear projections on the $k$ bottleneck slices, leading to slicing complexity of $\mathcal{O}(k(d+L))$ compared to $\mathcal{O}(dL)$ of the SW distance. The need for such hierarchical slices comes from deep learning applications where the number of samples $n$ (in mini-batch processing) is often smaller than their dimensionality $d$, and the computational complexity of the SW distance is dominated by the projection complexity. The more interesting part of this paper is the definition of Overparameterized Radon (OR) transform and its combination with the Partial Radon (PR) transform that results in the proposed Hierarchical Radon transform used to show that the proposed HSW is indeed a metric. Lastly, the authors demonstrate the application of HSW and compare it with SW for generative modeling on CIFAR10, CelebA, and Tiny ImageNet. They show boosts over FID and IS for different numbers of slices $L$ while demonstrating that HSW is faster than SW.

**Summary Of The Review:**

Overall, I think the paper is well-written, and it addresses a valid challenge for the application of sliced-Wasserstein distance in training neural networks when using small mini-batches. The proposed method is algorithmically simple, yet it is mathematically rigorous, and it provides a significant improvement over the SW distance.

---

> ### Author Response · Authors · 2022-11-12
> **Response to Reviewer EUsv**
>
> Thank you for your time and your feedback. We have revised the paper based on your suggestions. All modifications are written in blue color.
>
> **Q9**: When defining the Max-HSW variation, the maximization could be performed on the k bottleneck slices (I find this very interesting), on the L linear combinations of the bottleneck slices, or jointly (as it is currently defined in the paper). It would have been exciting to see the effect of only maximizing over the k bottleneck slices. Moreover, it seems that one should require the k bottleneck slicers to be linearly independent; what enforces this constraint?
>
> **A9**: Thank you for your insightful suggestion. In our Max-HSW, we do not enforce any linear independence constraint over $k$ bottleneck projecting directions.  However, that kind of constraint can be done by forcing the $k$ vectors to be orthogonal. Formally, we could project $k$ vectors into the Stiefel manifold by performing the Gram-Schmidt process. We have added this discussion to our revision in Appendix B. For the variant that only maximizes over the k bottleneck slices and taking the expectation over the linear mixing vector, we denote this variant as Semi max hierarchical sliced Wasserstein distance. Here, the optimization problem can be solved by doing a stochastic ascent algorithm. We have added this variant to our revision in Definition 9 in Appendix B. It is worth noting that we could also take the expectation over $k$ bottleneck projecting directions and maximize the linear mixing vector. Due to a large number of extensions of HSW including using non-linear mapping, augmenting mapping, and semi-maximization, we will leave the careful investigation of those variants to future work.
>
> **Q10**:  As far as I can see, if any of the projections (i.e., the  k bottleneck or the L  following slices) become nonlinear, then the metricity of HGSW would also depend on the defining functions and the injectivity of the (in this case) generalized hierarchical Radon transform. Could you please clarify this?
>
> **A10**: We have added detailed definitions of Overparametrized GRT, Partial GRT, and Hierarchical GRT in Definitions 10-12. To guarantee injectivity, the defining function must also satisfy some constraints (H1-H4 in [1]). For example, we could use the circular defining function and the polynomial of odd degrees. In summary, the metricity of HGSW also depends on the defining functions of GRT. We have revised our discussion accordingly in the revision.
>
> [1] Generalized Sliced Wasserstein distances, Kolouri et al.
>
> **Q11**: Propositions 1 and 2
>
> **A11**: Yes, HRT and ORT functions need to be the same for all arguments to guarantee injectivity. We have modified Propositions 1 and 2 based on our comments in blue in our revision.
>
> **Q12**: Typos
>
> **A12**: We have fixed all mentioned typos in our revision.

---

> > ### Comment · Reviewer_EUsv · 2022-11-15
> > **Response to the authors**
> >
> > I thank the authors for their response and their clarifications. I also reviewed my fellow reviewers' comments and the changes in the paper, and I think the changes have increased the paper's clarity.
> >
> > I agree that there is a large number of extensions of HSW (e.g.,  non-linear mapping, augmenting mapping, and semi-maximization), and testing all possible combinations can go well beyond the scope of this work. However, I think some of these extensions could provide significant performance boosts and could be studied in future work.
> >
> > I was very positive about this paper, and my view has not changed. So I still vote for its acceptance.

---

> > > ### Author Response · Authors · 2022-11-17
> > > **Response to Reviewer EUsv**
> > >
> > > Dear Reviewer EUsv,
> > >
> > > Thank you for your time and your constructive feedback. We will keep revising the paper based on the discussion with other reviewers.
> > >
> > > Best regards,
> > >
> > > Authors,

---

### Official Review · Reviewer_CrfW · 2022-10-24

**Confidence:** 4
**Correctness:** 4
**Technical Novelty And Significance:** 3
**Empirical Novelty And Significance:** 2
**Recommendation:** 5

**Clarity, Quality, Novelty And Reproducibility:**

The paper is well written and easy to read. The proposed method is novel and clearly explained.

Questions:
(1) How are the computational complexity and project complexity are computed? What are the units of those complexities as reported in Tables 1 and 2?
(2) The proposed HSW is constructed by recursively applying PRT and ORT to compared measures, is it possible to stack more PRTs and ORTs to construct SWD variants in the same way as in the HSW? Would they still be distance metrics? And what are their computational complexities compared to the HSW?
(3) What is the performance of the HSW when H>1?


**Strength And Weaknesses:**

Strength:

Theoretical properties of the HSW are derived in the paper, including its metricity, its link to SWD, max-SWD, approximation error of HSW, and computational complexity. Compared with the standard SW distance, the proposed HSW has the advantage of lower computational cost when both the data dimension and the number of final projections are much larger than the number of bottleneck projections k.

The proposed HSW is evaluated on a benchmark deep generative modelling experiments to compare it with other variants of SWDs. Experiment results show that models trained with HSW can produce images of higher qualities in terms of FID and IS scores. The HSW also demonstrated higher convergence rate with approximately the same computational costs than SWD.

Weakness:

My major concern is on the experiment side. The main context of the proposed work is to improve computational efficiency of the optimal transport-based distances, in particular sliced-based ones, but the proposed HSW is only compared with the standard SWD, ignoring most of the recent efforts which were also discussed in the paper. The proposed method is also only evaluated on a generative modelling experiment, but other applications of slice-based Wasserstein distances can be found in other related papers, e.g., colour transferring and sliced iterative normalising flows [1][2].

[1] Soheil Kolouri, et al. "Generalized Sliced Wasserstein Distances." NIPS, 2019.
[2] Biwei Dai, and Uros Seljak. "Sliced Iterative Normalizing Flows." ICML, 2021.

**Summary Of The Paper:**

The paper introduced a new variant of sliced Wasserstein (SW) distance, the hierarchical sliced Wasserstein (HSW) distance. The HSW distance is derived based on the proposed hierarchical Radon transform (HRT), which is a composition of partial Radon transform and overparametrised Radon transform. In the HSW, compared probability measures are first projected onto k bottleneck directions, yielding new probability measures $f_\mu(t_{1:k}, \theta_{1:k})$ and $f_\nu(t_{1:k}, \theta_{1:k})$, then the generated measure are projected onto a one-dimensional space using the partial Radon transform (PRT) in a $k$-dimensional subspace with L directions $\psi$. The HSW is defined as the expectation of Wasserstein distances between the obtained one-dimensional probability measures over the distribution of $\theta_{1:k}$, a product probability measure of k uniform distributions on d-dimensional unit spheres, and the distribution of $\psi$, a uniform distribution on a k-dimensional unit sphere. In short, final projections in the HSW are linear combinations of the k bottleneck projections, and the weights of the k projections are given by L random samples drawn from a uniform distribution on a k-dimensional unit sphere.

**Summary Of The Review:**

The paper can be improved if more comprehensive experiments are included to support the effectiveness of the proposed method.

---

> ### Author Response · Authors · 2022-11-12
> **Response to Reviewer CrfW - Part 1**
>
> We appreciate your time and your comments. We have revised the paper based on your suggestions. All modifications are written in blue color.
>
> **Q4**: the proposed HSW is only compared with the standard SWD.
>
> **A4**: The main message that we want to convey in the paper is that the hierarchical construction of projections could reduce computational costs. The reason that we compare HSW with only SW is for keeping the simplicity of the comparison between the conventional approach and the hierarchical approach. As discussed in the paper, the hierarchical approach can be further applied to non-linear projections via Generalized Radon Transform, augmenting sliced Wasserstein and distributional sliced Wasserstein as in the discussion with Reviewer XSPt (Q2).  We have added these extensions to the revision in Appendix B. Therefore, the contribution of the paper is orthogonal to the contributions of previous works. Again, we would like to note that the purpose here is not to ignore the effort of previous works but to bring out the computational aspect of an important baseline. We have shown our appreciation for the suggested papers by adding careful discussions in the revision.
>
> **Q5**: The proposed method is also only evaluated on a generative modeling experiment, but other applications of slice-based Wasserstein distances can be found in other related papers, e.g., colour transferring and sliced iterative normalising flows.
>
> **A5**: We agree that there are a lot of applications where HSW can be applied including color transfer, domain adaptation, autoencoder, point cloud reconstruction, and so on. In the paper, we focus on the setting that the number of supports $n$ is smaller than the number of dimensions $d$ ($n<<d$) to show the computational benefit of the hierarchical approach via bottleneck projections. Therefore, the deep generative modeling application is the best fit for that purpose. In other applications, where $d$ is small and $n$ is large , e.g., color transfer (d=3, n=3000), HSW does not offer a computational advantage. However, it could offer a new and dependent structure between final projecting directions. We have added additional experiments on color transfer in Figure 8 in Appendix E in the revision. We compare SW with HSW, and Generalized sliced Wasserstein (GSW) with Hierarchical Generalized sliced Wasserstein (HGSW) using the polynomial of degree 3 defining function. We show that the hierarchical approach gives better color-transferred images (measured by the Wasserstein-2 scores between the color-transferred images' color palette and the color palette of the target image). However, the downside is that the hierarchical approach has a computational time only comparable to the traditional approach, namely, the hierarchical approach requires slightly more computation.
>
> For the sliced iterative normalizing flows, the paper proposes a new generative modeling framework by utilizing one-dimensional transportation maps. The source distribution is iteratively moved to the target distribution. In contrast, our generative modeling is based on parameter estimation with a minimum expected distance estimator [1]. To the best of our knowledge, the parameter estimation approach gives better FID and IS scores. We have included the FID scores and IS scores from the conventional sliced iterative normalizing flows in Table 1. It is worth noting that sliced iterative normalizing flows also use Radon Transform, hence, Hierarchical Radon Transform can be potentially applied to the framework. We leave this investigation for future work.
>
> [1] Asymptotic Guarantees for Learning Generative Models with the Sliced-Wasserstein Distance, Nadjahi et al.
>
> **Q6**: How is the computational complexity and project complexity computed? What are the units of those complexities as reported in Tables 1 and 2?
>
> **A6**: In Table 1 of the paper, we report the computational complexity and the projection complexity of SW computed by $(Ln\log n + Ldn)*10^{-6}$ and $(Ld)*10^{-6}$. Also, the computational complexity and the projection complexity of HSW are computed by $(Ln\log n + kdn+Lkn)*10^{-6}$ and $(kd+Lk)*10^{-6}$. The numbers are representatives of the scaling complexities of SW and HSW. We have added an explanation to the revision in blue.

---

> > ### Comment · Reviewer_CrfW · 2022-11-21
> > **After rebuttal**
> >
> > I thank the authors for the response. I am satisfied with A6-A8. For A4 and A5, and in light of A2 to Reviewer XSPt, I think that adding comments about connections with those approaches as in Appendix B, do not really address Q4 and Q5, particularly Q4. A description of variants introduced in Appendix B without experiment validation can make the paper slightly incomplete to me and can reduce the impact of the work. So, I keep my score unchanged.

---

> > > ### Author Response · Authors · 2022-11-22
> > > **Response to Reviewer CrfW**
> > >
> > > Thank you for your response. We appreciate your time and feedback.
> > >
> > > We first would like to recall that we have conducted experiments on color transfer where we run the application of the hierarchical projection approach to generalized sliced Wasserstein distance (GSW) which is hierarchical generalized sliced Wasserstein distance (HGSW). Moreover, we have also added the result of the sliced iterative normalizing flow to Table 1.
> > >
> > > Secondly, the main contribution of our paper is the hierarchical projection approach which is the hierarchical Radon Transform. In the paper, we focus on comparing the proposed hierarchical projection approach to the conventional projection approach in the form of hierarchical sliced Wasserstein distance (HSW) and sliced Wassersteind distance (SW). We believe that the hierarchical projection approach is orthogonal to the projection selection approach (distributional sliced Wasserstein distance) and the augmenting approach (augmented sliced Wasserstein distance). As evidence, we have included some potential definitions of the intersection of the hierarchical approach to those approaches. We believe that each intersection deserves careful work to investigate its theoretical properties and empirical performance.
> > >
> > > Overall, we would be grateful if you could reevaluate our paper based on the potential of the hierarchical projection approach.
> > >
> > > Best regards,

---

> ### Author Response · Authors · 2022-11-12
> **Response to Reviewer CrfW - Part 2**
>
> **Q7**: The proposed HSW is constructed by recursively applying PRT and ORT to compared measures, is it possible to stack more PRTs and ORTs to construct SWD variants in the same way as in the HSW? Would they still be distance metrics? And what are their computational complexities compared to the HSW?
>
> **A7**: Thank you for an insightful question. We have discussed the usage of more than a single level of hierarchy in Section 3.2. More formally, we could use more than one bottleneck layer, e.g., $k_1,k_2,\ldots,k_N, L$ with $k_1, k_2,\ldots, k_N <  L$ ($N \geq 2$). We denote this variant as N-$HSW$. The computational complexity of this variant is $\mathcal{O}(Ln\log n + k_1dn +k_2 k_1n + \ldots, k_N k_{N-1} n + Lk_N n) $ and the projection complexity is $\mathcal{O}( k_1d +k_2 k_1 + \ldots + k_N k_{N-1}+ L k_N)$. As long as $k_1 d Lk_N + \sum_{i=2}^N k_i k_{i-1} < Ld$, N-$HSW$ is faster and consume less memory than SW. N-$HSW$ is still a metric due to the injectivity of the transformation in each layer. However, from the computational point of view, there is no benefit to using more than one bottleneck layer. Namely, we can simply choose $k = \min{k_1,\ldots,k_N\}$, then HSW with $k$ bottleneck projections will be more efficient than N-$HSW$. However, increasing the number of Radon Transform variants layer could add more structure and dependency to the final projecting directions. Moreover, using multiple layers could create a more complex non-linear function in the case of the Hierarchical Generalized Radon Transform (HGRT) by stacking multiple injective ones. We refer the reviewer to Appendix B for the definition of HGRT.
>
> In summary, as long as the bottleneck structure is used, using the hierarchical approach could improve the computation compared to the conventional approach. Compared to a single level of hierarchy (ORT + PRT), using more layers of Radon Transform variants does not have any computational benefits. However, using multiple layers could offer other potential benefits such as the structure and dependency between the final projecting directions and creating a more complex non-linear mapping via Generalized Radon Transform.
>
> **Q8**: What is the performance of the HSW when H>1?
>
> **A8**: $H$ plays a similar role as $L$ in the empirical HSW which is the number of Monte Carlo samples to approximate the expectation in the population HSW. We have added a theoretical statement about the Monte Carlo approximation error in HSW in Proposition 5 in Appendix B. In summary, the error is proportional to the inverse square root of $H$ and $L$. Therefore, for the same value of $L$, increasing $H$ leads to a better approximation performance. In the paper, we set $H=1$ to explain the usage of bottleneck projecting directions. Namely, the final projections of HSW are simply created by performing two matrix multiplications.

---

> ### Author Response · Authors · 2022-11-21
> **Look forward to your feedback**
>
> Dear Reviewer CrfW,
>
> We have addressed your concerns in our responses. Given that you are the only one that gives a negative score on our paper, we would like to hear your feedback. Please feel free to raise questions if you have other concerns.
>
> Best regards,
>
> Authors

---

### Official Review · Reviewer_XSPt · 2022-10-26

**Confidence:** 4
**Correctness:** 4
**Technical Novelty And Significance:** 3
**Empirical Novelty And Significance:** 3
**Recommendation:** 6

**Clarity, Quality, Novelty And Reproducibility:**

The code is attached to the supplementary materials. I thank the authors for that. Reproducibility of results is guaranteed.
The paper is easy to follow and well-explained.

### Typos ###
- Page 1: "sample complexity": it is a little bit confusing, since this term we use for the number of training samples that we need to supply to a learner algorithm. It will be better if the authors clarify this point as the bounding gap between the evaluation of divergence on two probability measures versus samples from these measures.
- Page 5: "success of overparametrization in DNN" (Add a reference or details).
- Page 7: "while comparing HSW with the SW.e experiments while comparing the HSW with the SW" --> "while comparing the HSW with the SW"
- References: "radon" --> "Radon"


**Strength And Weaknesses:**

### Strength ###
- HWD: novel variant of sliced Wasserstein distance based on a hierarchal Radon transform.
- HWD is a proper distance in the space of probability measures.
- Equivalence between HWD, SWD, and Max-SWD.
- Numerical experiments on generative modeling with HWD.

### Weaknesses ###
- In most experiments, the number of projections $L$  is taken significantly large. So, this affects badly the computational cost of calculating the linear mixing of the bottleneck projections. This mean, at last for me, HWD gains efficiency with a price of a large number of projections. At that time, one can consider vanilla SWD. I will appreciate it if the authors give more details about this point.
- The paper lacks comparison with other approaches like augmented sliced Wasserstein (Chen et al. ICLR 2022) and distributional sliced Wasserstein (Nguyen et al. ICLR 2021).

**Summary Of The Paper:**

Applications of Wasserstein distance to large-scale machine learning problems have been limited by its enormous computational cost. The Sliced Wasserstein (SW) distance and its variants increase computational efficiency using random projections but suffer from low accuracy if the number of projections is not large enough. In this work, the authors propose a new family of sliced-Wasserstein distance measures, called Hierarchical Sliced Wasserstein distances (HSWDs).

HSWDs are based on projecting original measures into $k$ one dimensional projected measure via Radon transform with $k\ll L$ and where $L$ is the number of projections. Towards this end, the authors define hierarchical Radon transform, which has an injectivity property. This latter yields the metricity property of HSWDs. Several numerical experiments in generative modeling are conducted on CIFAR10, CelebA and Tiny ImageNet datasets.

**Summary Of The Review:**

the paper introduces novel variants of sliced Wasserstein distance to reduce the computational cost. The paper lacks comparison with other approaches like augmented sliced and/or distributional sliced Wasserstein distances.

---

> ### Author Response · Authors · 2022-11-12
> **Response to Reviewer XSPt**
>
> Thank you for your time and your feedback. We have revised the paper based on your suggestions. All modifications are written in blue color.
>
> **Q1**: In most experiments, the number of projections L is taken significantly large. So, this affects badly the computational cost of calculating the linear mixing of the bottleneck projections. This mean, at last for me, HWD gains efficiency with a price of a large number of projections. At that time, one can consider vanilla SWD. I will appreciate it if the authors give more details about this point.
>
> **A1**: Thanks for your comment. We would like to recall that the computational complexity of SW is $\mathcal{O}(Ln\log n + Ldn)$ and the projection complexity of SW is $\mathcal{O}(Ld)$ where $n$ is the number of supports, and $d$ is the number of dimensions. The computational complexity of HSW is $\mathcal{O}(Ln\log n + kdn + Lkn)$ and the memory complexity of HSW is $\mathcal{O}(kd+Lk)$. In the paper, we focus on the setting where $n <<d$ ,i.e., the number of supports is smaller than the number of dimensions. In particular, in the deep generative model application, $n=128$ and $d=8912$. Therefore, $Ln\log n << Ldn $ implies that the computation for projecting measures is the main computational bottleneck. HSW decomposes the projection matrix of size $d \times L$ into two matrices of size $d \times k$ and $k\times L$ with $k<L$. Hence, for the same computational complexity, HSW can use a bigger $L$ compared to the SW. In Table 1 of the paper, we report the computational complexity and the projection complexity of SW computed by $(Ln\log n + Ldn)*10^{-6}$ and $(Ld)*10^{-6}$. Also, the computational complexity and the projection complexity of HSW are computed by $(Ln\log n + kdn+Lkn)*10^{-6}$ and $(kd+Lk)*10^{-6}$. From the table, we observe that for less computation and memory, HSW gives better results than SW due to the usage of more final projections. This is the main benefit of using the proposed Hierarchical Radon Transform: creating more final projections with less computation.
>
> **Q2**: The paper lacks comparison with other approaches like augmented sliced Wasserstein (Chen et al. ICLR 2022) and distributional sliced Wasserstein (Nguyen et al. ICLR 2021).
>
> **A2**: Thank you for your recommendation. Augmented sliced Wasserstein (ASW) (Chen et al. ICLR 2022) proposes a Spatial Radon Transform that first transforms the original measures of $d$ dimension to a higher dimension measures $2d$ by a learnable map. After that ASW uses Radon Transform to map the transformed measures to one-dimension projections. The benefit of ASW is to make two original measures linear-separable after transformation to higher dimensions.  However, the downside of ASW is increasing computational complexity and projection complexity due to doubling the dimension. Moreover, an iterative loop is required to learn the mapping in ASW. In contrast, HSW only uses random projections and keeps the dimension unchanged. It is worth mentioning that our proposed hierarchical Radon transform can also be applied to ASW by replacing Radon Transform after the transformation step. We refer to this variant as Spatial Hierarchical Radon Transform. Moreover, we can also use the transformation step in ASW to both Overparameterized  Radon Transform and Partial Radon Transform to create a new variant, Hierarchical Spatial Radon Transform. We have added these variants to our revision in Definition 15-16 in Appendix B.
>
> In distributional sliced Wasserstein (DSW), the uniform distribution for sampling projecting directions is replaced by a learnable distribution that can maximize the expected projected distance. Like ASW, DSW also requires a stochastic gradient ascent algorithm to find the best projecting distribution which is computationally expensive. In HSW, we still use the uniform distribution over the bottleneck projecting directions and the linear mixing weights. We can change the uniform distributions to better distributions like in DSW to obtain distributional hierarchical sliced Wasserstein distance. However, that comes at the expense of additional computation. We have added this variant to our revision in Definition 14 in Appendix B.
>
> To summarize, the main contribution of HSW is the hierarchical approach to create projections. This approach reduces the computation and potentially benefits the quality of projections. As suggested by the reviewer, we have added the discussion above to our revision. We believe the contribution is orthogonal to the augmentation approach and the distributional approach.  Due to the large portion of work, we leave the careful investigation of the distributional hierarchical sliced Wasserstein distance and spatial hierarchical sliced Wasserstein distance to future works.
>
> **Q3**: Typos
>
> **A3**: We have fixed all mentioned typos based on your comment.

---

> > ### Comment · Reviewer_XSPt · 2022-11-19
> > **Thank you for the explanation**
> >
> > I thank the authors for their thorough responses to my questions and concerns.

---

> ### Author Response · Authors · 2022-11-18
> **Look forward to your feedback.**
>
> Dear Reviewer XSPt,
>
> We have addressed your concerns in our responses.  We would like to hear your feedback since the deadline is approaching. Please feel free to raise questions if you have other concerns.
>
> Best regards,
>
> Authors

---

### Decision · Program_Chairs · 2023-01-20

**Decision:**

Accept: poster

**Justification For Why Not Higher Score:**

* lack of  large-scale/in-depth empirical evaluations

**Justification For Why Not Lower Score:**

the paper proposes a novel and interesting contribution relevant to the ML community

**Metareview: Summary, Strengths And Weaknesses:**

The paper introduces an extension of the sliced wasserstein distance that has reduced complexity.  Authors analysed the theoretical properties of this novel distance by introducing extensions of the Radon transform.
Experimental results confirm the usefulness of the proposed approaches but I may have been helpful to have an in-depth comparisons (see reviewers comment).

 Most reviewers agreed that the paper is interesting and relevant and the revised version of the paper added some experiments related to
color transfer which improved the experimental part. However, the paper would benefit from further empirical evaluation
and comparison with plain SWD in other practical applications.

**Note From Pc:**

if the above contains the word "oral" or "spotlight" please see: "oral" presentation means -> notable-top-5% and "spotlight" means -> notable-top-25%. As stated in our emails, we are disassociating presentation type from AC recommendations